# DIS3 mutations enhance AID-driven translocations during B-cell activation, promoting transformation to multiple myeloma

DIS3, a key nuclear RNA-degrading enzyme, is essential for immunoglobulin class switch recombination (CSR), promoting activation-induced cytidine deaminase (AID) activity on both DNA strands to induce double-strand DNA breaks. During somatic hypermutation, AID-dependent lesions predominantly occur on the non-template DNA strand. Dominant mutations impairing DIS3 exoribonucleolytic activity are common in multiple myeloma (MM), but their role in carcinogenesis remains unclear. Here we show, using a knock-in mouse model, that the clinically relevant DIS3 G766R variant causes chromosomal translocations in B-cells, characterized by aberrant AID activity signatures. The mice develop pristane-induced plasmacytomas, modeling early-stage MM. In clinical MM samples, DIS3 mutations correlate with *IGH* translocations and AID-driven lesions in driver genes. Mechanistically, mutated DIS3 accumulates on chromatin-bound RNA, particularly at aberrant AID target sites, promoting mutations on both DNA strands. This results in increased AID-dependent double-strand DNA breaks, fostering microhomology-mediated oncogenic rearrangements. Translocations occur specifically during CSR, which remains functionally intact. The DIS3 G766R mutation does not disrupt chromatin architecture in activated B cells but exploits spatial proximity to permanently juxtapose enhancers and proto-oncogenes, facilitating transformation. Thus, gain-of-function DIS3 mutations enhance AID promiscuity, driving IGH translocations and MM development without broadly affecting B-cell physiology.

*DIS3* is a critical gene for the function of every cell, which at the same time is frequently mutated in multiple myeloma (MM)[1-3]. It encodes a catalytic subunit of the eukaryotic nuclear exosome complex, a ribonuclease involved in the processing of stable RNA species, such as ribosomal RNA (rRNA) and degradation of cryptic transcription products, including promoter upstream transcripts (PROMPTs) and enhancer RNA (eRNA)[4-6]. Most prominently, DIS3 degrades products of spurious RNA polymerase II activity: low-level, unannotated, cryptic transcription that can account for up to 70% of the genome[6]. This

activity links the nuclear exosome to the regulation of super-enhancer activity, in part by affecting chromatin topology[7].

The specific role of the nuclear exosome, particularly DIS3, in B-cells is associated with immunoglobulin diversification via class switch recombination (CSR) and somatic hypermutation (SHM)[8-10], processes involving a large number of trans-acting factors. During maturation of B cells, activation-induced cytidine deaminase (AID) deaminates cytosine residues in DNA, leading to the production of uracil and generating G:U mismatches that recruit DNA repair

e-mail: tkulinski@iimcb.gov.pl; adziembowski@iimcb.gov.pl

machinery. Uracil bases are then excised by the repair enzyme uracil-DNA glycosylase, resulting in abasic sites. This is followed by the cleavage of the DNA backbone by apurinic endonuclease, which creates single-strand breaks, that can be filled in by error-prone DNA polymerases, resulting in point mutations or indels (in SHM). Mismatches on both DNA strands lead to staggered double-stranded DNA breaks, repaired by classical non-homologous end joining (c-NHEJ) or microhomology mediated alternative end joining pathway (a-EJ), resulting in large deletions that constitute CSR[11,12]. Specific DNA targeting mechanisms ensure that AID activity is largely restricted to the immunoglobulin heavy chain locus (*IGH*) where SHM and CSR take place[13,14]. It involves transcription and rapid RNA degradation[15]. Target regions are generally GC-rich, contain tandemly repeated DGYW DNA sequence motifs, have the potential to form G-quadruplexes, and are prone to RNA polymerase stalling and to RNA-DNA hybrids (R-loops) formation[16–22]. In vitro, AID predominantly targets the non-template strand during transcription. In vivo, AID mutates both DNA strands, necessitating a special mechanism involving the DIS3/exosome to displace the nascent RNA[9,23]. The full mechanistic details are not yet known. Nevertheless, tight regulation of AID is critical, as its off-target activity is well-known to cause genetic lesions leading to B-cell lineage cancers[24–28]. Interestingly, super-enhancer regions in B-cells, which possess most of the features required by AID, are critical sites for AID recruitment and serve as hotspots for AID-mediated genomic instability[29].

The *Igh* locus forms a topologically associating domain (TAD), which tightly regulates both the transcriptional activity of regions undergoing switch recombination and promotes deletional CSR in *cis*. In naïve B-cells, loop extrusion juxtaposes the 3' *Igh* regulatory region (3' RR) enhancers with the Sμ region. B-cell activation, transcription and the loading of the cohesin complex result in the generation of dynamic subdomains that directionally align the donor Sμ region with one of the downstream S regions for deletional CSR[30,31]. Upon *DIS3* KO, the chromosome architecture is abrogated, and CSR is lost[8], adding another layer of complexity to the DIS3 function in B-cells.

MM is the second most prevalent hematological malignancy among adults. It originates during the terminal differentiation of B lymphocytes into plasma cells, which reside in the bone marrow and produce massive amounts of antibodies[32,33]. MM cells, like plasma cells, have already undergone all genome editing steps associated with the maturation of activated B lymphocytes, including CSR and SHM[20,34]. MM is classified based on its genetic lesions into hyperdiploid (HRD), distinguished by trisomies of several chromosomes, and non-hyperdiploid (non-HRD), characterized by chromosomal translocations[32,35]. Many translocations in MM involve the *IGH* locus, leading to the overexpression of oncogenes such as *FGFR3, MMSET (NSD2), CCND3, CCND1, MYC*, and *MAF*[2,3,32,35,36]. A murine pristane-induced peritoneal plasmacytoma (PCT) models the occurrence of translocations driving neoplastic transformation as observed in MM[37]. In the BALB/c strain of mice, these chromosomal alterations typically originate at the *Igh* locus, directly or indirectly leading to the upregulation of the *Myc* locus[38–40]. These events are strictly dependent on the promiscuous activity of AID[24,27]. Furthermore, clinical data clearly indicate that AID-induced point mutations in a subset of MM driver genes are crucial for the development of human MM[41].

Ten to 20% of MM cases harbor somatic point mutations in the DIS3 gene. These mutations are specific to non-HRD MM, and, notably, DIS3 is the only RNA exosome subunit associated with cancer[2,42,43]. Exosome dysfunction typically disrupts post-transcriptional regulation of gene expression, leading to reduced cell proliferation and increased cell death[5,44–47]. At the same time, exosome depletion or broader dysfunctions in nuclear RNA 3' end processing[4,48–56] often exhibit mutagenic potential in various models. In this context, it was recently shown that oncogenic mutations in one of the cyclin-dependent protein kinases (CDK13) impair the function of cofactors

of the nuclear RNA exosome, specifically ZC3H18 and SKIV2L2, resulting in deficiencies in nuclear RNA surveillance and RNA exosome activity[57]. Notably, MM-associated DIS3 mutations are very specific to the protein regions responsible for its exoribonucleolytic activity. These mutations inhibit or, in the case of recurrent mutations (D479, D488, and R780), completely abolish the exoribonucleolytic activity of this enzyme, exerting dominant-negative effects[58]. Furthermore, these mutations arise early in MM progression[42,59], presumably driving the disease[2,3,60]. However, they are counter-selected in later stages of MM due to their inhibitory effects on cell proliferation[58].

Analyses of clinical samples and MM cell lines did not reveal the exact oncogenic mechanism of *DIS3* gene mutation. Conflicting conclusions were drawn on whether this mechanism arises as a consequence of transcriptome deregulation[43] or not[58]. Studies in cellular models, which suggested a generalized genome destabilization effect, primarily used DIS3 knockdown (KD) models and cell lines outside the B-cell lineage, failing to replicate the fact that only point mutations, rather than complete knockdowns, have been observed in MM patients[61]. Our attempts to generate mouse lines with recurrent MM *DIS3* alleles failed because of strong dominant-negative effects, leading to early embryonic lethality[58]. Therefore, to study the mechanistic implications of DIS3 dysfunctions in the development of B cell tumors, we decided to focus on a milder, yet clinically relevant DIS3 variant with glycine 766 mutated to arginine (DIS3[G766R])[3]. DIS3[G766R] is predicted to be highly pathogenic (Supplementary Fig. S1a), and previous experimental validation revealed that on a protein level, it leads to DIS3 protein stalling on structured RNA substrates[5,62]. Still, single-stranded RNA can be degraded[5,62].

Herein, we generate a DIS3[G766R] knock-in mouse model to further our understanding of the effect of clinical DIS3 variant on B-cell development, antibody maturation, and DIS3-mediated carcinogenesis in MM. Our analysis, supported by genomic studies of clinical samples, allows us to propose a model whereby the DIS3 mutations lead to aberrant accumulation of DIS3 at sites of AID activity. The presence of DIS3 at these sites drives oncogenic chromosomal translocations. In contrast to DIS3 loss-of-function mutations, however, MM-associated DIS3 variants do not abrogate CSR or affect genome architecture.

## Results

### Heterozygous MM-associated *Dis3*[G766R/+] mutation leads to subtle deregulation of non-coding regions of the genome

To study the role of *DIS3*[G766R] in a native genomic context, we introduced the G766R in the mouse *Dis3* locus in the C57BL/6 background (Supplementary Fig. S1b). Similar to the complete *Dis3* knockout, the homozygous Dis3[G766R] mutation is embryonically lethal (Supplementary Fig. S1c) in agreement with the pathogenic character of this clinically observed DIS3 mutation. At the same time, heterozygous *Dis3*[G766R] animals did not exhibit developmental phenotypes nor hematologic abnormalities (Supplementary Fig. S1d,e).

As a next step, we profiled the transcriptomes and proteomes of in vitro-activated WT and *Dis3*[G766R/+] B-cells isolated from mice. Consistent with the lack of developmental phenotypes, the effect of *Dis3*[G766R/+] on mRNA levels was very subtle, contrasting the massive gene expression change and activation of genes related to DNA damage response upon DIS3 inactivation[5,58,61,62]. We found 16 differentially expressed genes (DEGs) in naïve, three DEGs in B cells upon 3 days of activation, and no DEGs in B-cells activated for 5 days (Fig. 1a, Supplementary Fig. S2a). Furthermore, no transcript was upregulated by more than 2-fold. We also analyzed transcriptional responses accompanying differentiation after in vitro activation of B-cells and observed no significant effect of heterozygous *Dis3*[G766R] mutation (Fig. 1b, Supplementary Fig. S2b). Accordingly, there were no significant changes at the proteome level of Dis3[G766R/+] and Dis3[+/+] primary B-cells activated for 3 days in vitro (Fig. 1c).

   

Given the limited effect of Dis3[G766R/+] on the steady-state transcriptome, we aimed to test whether RNA/DNA hybrids accumulate in exosome-deficient cells, potentially destabilizing the genome. For this reason, we used DNA/RNA hybrid immunoprecipitation coupled with high-throughput RNA-sequencing (DRIP-seq). Comparing the DRIP-seq over the exons of the annotated transcriptome of *Dis3[G766R/+]* and *Dis3[+/+]* B-cells, we scored virtually no effect of the mutation (Fig. 1d).

Major DIS3 substrates represent non-protein coding RNA species with massive (up to 100-fold) accumulation of PROMPTs at the steady state of DIS3 dysfunctional cells[6]. In the case of Dis3[G766R/+] B-cells compared to WT, no accumulation was detected when looking at individual ncRNA species. However, global analysis of the RNA-seq signal centered over active transcriptional start sites (TSS) revealed focal enrichments in regions corresponding to RNA Pol II promoter-proximal pausing and upstream regions from which PROMPTs originate (Fig. 1e and Supplementary Fig. S2d, e). Notably, in *Dis3[G766R/+]* B-cells, DRIP-seq analysis revealed a much more prominent accumulation of R-loops in the promoter-proximal pausing regions and over PROMPTs (Fig. 1f). Although R-loops-engaged RNA represent only a

fraction of DIS3 substrates, their highly transient nature and the sensitivity of DRIP-seq allow for their effective detection. This indicates a transient yet detectable impact on the transcriptome, consistent with DIS3's known role. Similar transcriptomic effects were also observed for embryonic fibroblasts (MEFs) (Supplementary Fig. S2f, g).

Interestingly, the subtle effects of the Dis3[G766R/+] on the transcriptome have some impact on the physiology of mice. In contrast to the heterozygous *Dis3* knockout mice, which do not differ in size from WT, *Dis3[G766R/+]* animals were smaller than their WT littermates (Fig. 1g, Supplementary Fig. S3a–d). This reduced size corresponds to a negative impact on cell proliferation both in activated B-cells and MEFs (Fig. 1h, Supplementary Fig. S3e, f), aligning with a conserved phenotype of DIS3 dysfunction, which includes reduced mitotic progression and increased cell death[5,44–47].

We conclude that primary B cells do not show any detectable defects on the annotated protein-coding transcriptome or proteome. Instead, the molecular phenotype is limited to unstable R-loop forming on non-protein-coding RNA species vulnerable to the biochemically mild heterozygous DIS3[G766R] variant.

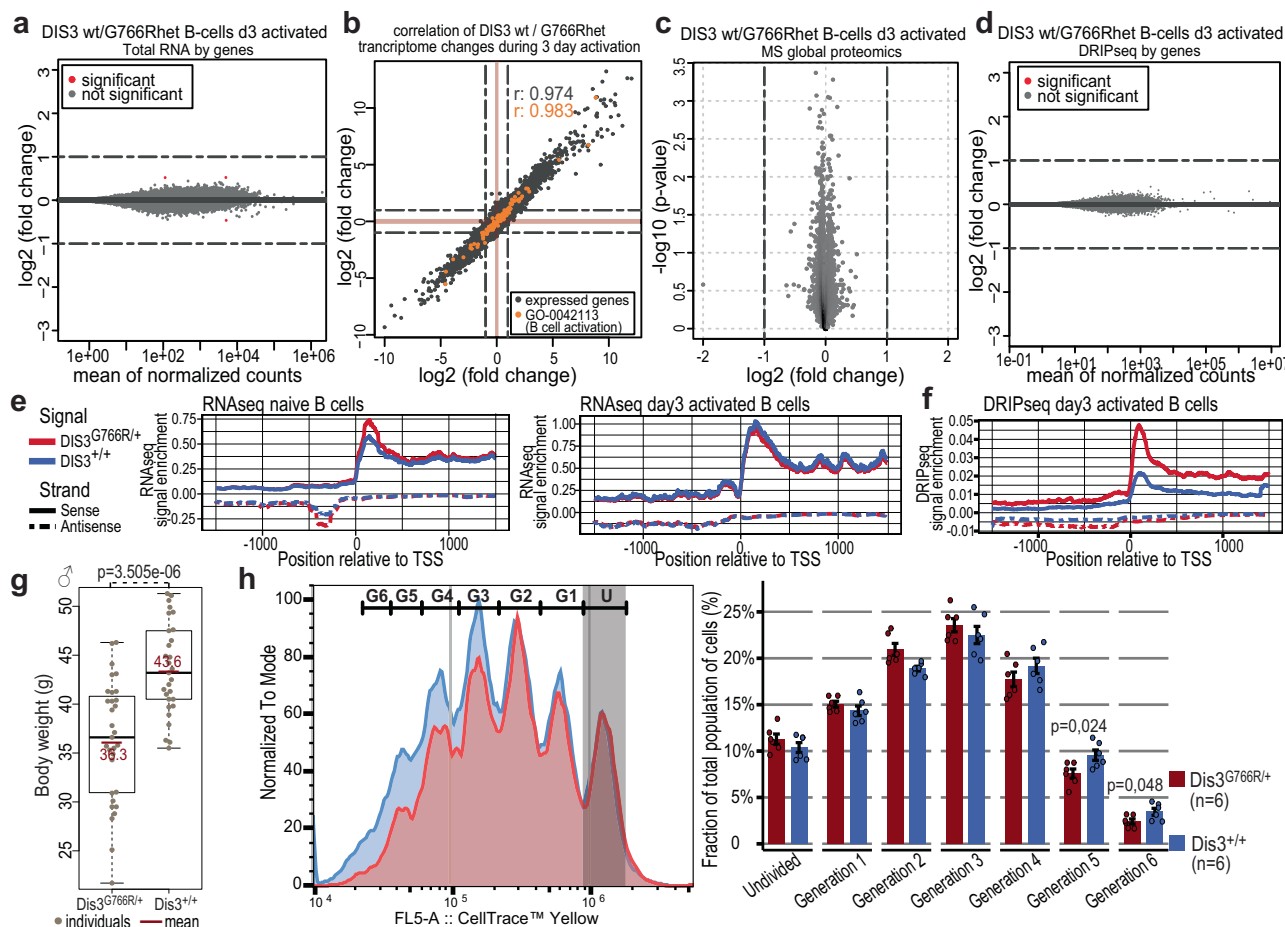

**Fig. 1 | Dis3[G766R/+] knock-in mice exhibit mild deregulation of non-coding regions. a** MA plot comparing the transcriptomes of primary Dis3[G766R/+] and Dis3[+/+] B-cells, in vitro activated for 3 days (*n* = 3 for each genotype). **b** Correlation of transcriptome changes in the DIS3[G766R/+] and DIS3[+/+] during in vitro activation for 3 days. **c** Global proteome MS analysis of Dis3[G766R/+] and Dis3[+/+] primary B-cells, in vitro activated for 3 days (*n* = 3 for each genotype). **d** MA plot comparing RNA/DNA hybrid DRIP-seq signal (with subtracted RNase H control signal) over the annotated transcriptome Dis3[G766R/+] and Dis3[+/+] B-cells, in vitro activated for 3 days (*n* = 2 for each genotype). Dashed lines on all plots represent a threshold of 2-fold change. (Statistics for all MA plots: two-sided Wald test with Benjamini–Hochberg multiple-testing correction.) **e** Meta-analyses of RNA-seq

signal over TSS of expressed genes in naïve and day 3 activated B-cells. **f** Meta-analyses of DRIP-seq signal over TSS of expressed genes in day 3 activated B-cells. **g** Body weight of male WT and heterozygous littermate Dis3[G766R/+] mice aged 30-50 weeks. (Box plots show the median (center line), 25th–75th percentiles (box), whiskers to 1.5×IQR, *n* = 30 individuals of each genotype plotted as points) **h** Representative examples of CellTrace™ Yellow proliferation assay performed on splenic B-cells activated in vitro for 3 days. Quantification of CellTrace™ Yellow proliferation assay performed on B-cells activated in vitro for 3 days, with 6 biological replicates per genotype. Statistical analysis was conducted using a two-tailed t-test. Error bars represent the standard deviation. Source data are provided as a Source Data file.

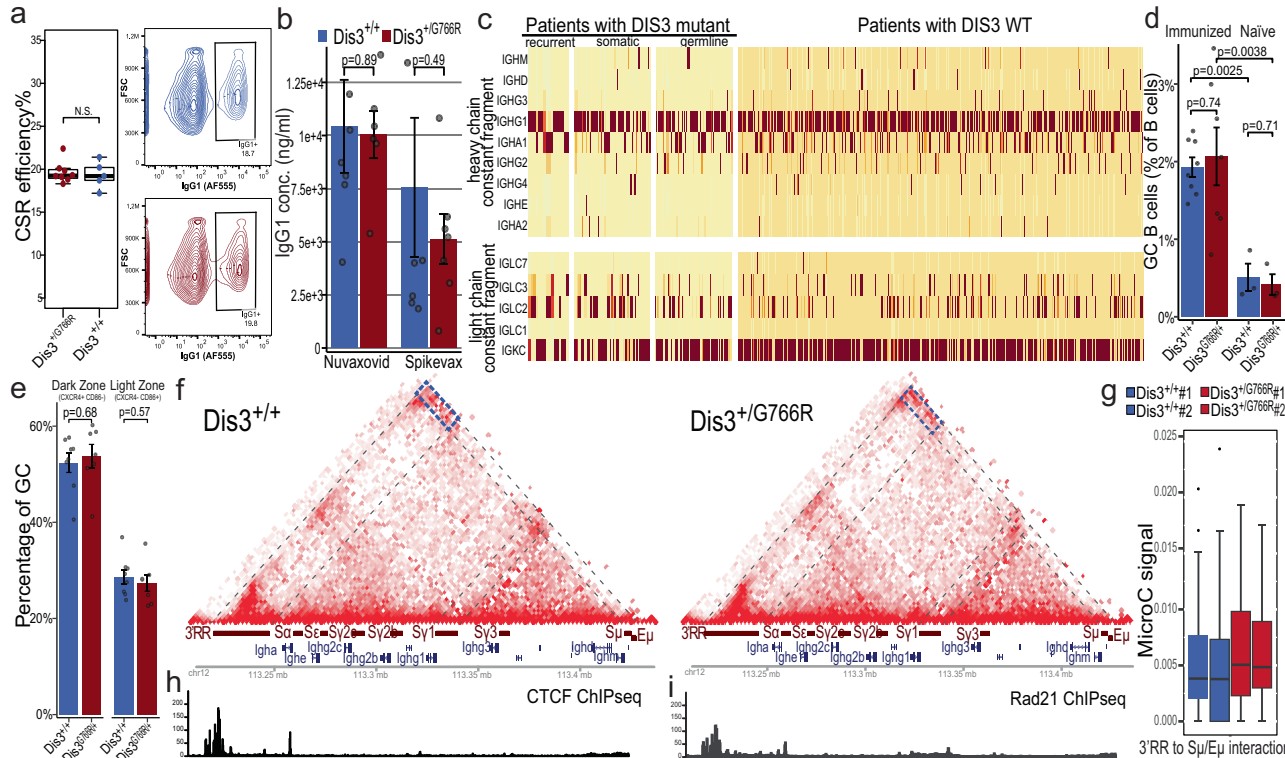

**Fig. 2 | Unaffected immunoglobulin class switch recombination in B and MM cells with DIS3 MM variants. a** No significant changes in frequencies of switching to the IgG1 class are observed in WT and Dis3$^{G766R/+}$ B-cells 3 days after activation with lipopolysaccharide and interleukin-4. (Box plots show the median (center line), 25–75th percentiles (box), whiskers to 1.5×IQR. $n = 8$ Dis3$^{G766R/+}$ and $n = 5$ Dis3$^{+/+}$ individuals) **b** ELISA of α-SPIKE IgG1 antibodies, 14 days after immunization of mice with Nuvaxovid and Spikevax vaccines. ($n = 7$ individuals per group, except Nuvaxovid Dis3$^{G766R/+}$ $n = 6$; Error bars represent SEM). **c** No significant changes in frequencies of class switching in WT DIS3 patients or patients with MM-specific DIS3 mutations, both somatic and germline, as determined by the analysis of immunoglobulin heavy and light chain classes expression in RNA-seq experiments. **d** Flow cytometric analysis revealed a significant expansion of splenic GC B cells (CD95$^+$ GL7$^+$) in immunized versus naïve mice, with a magnitude comparable between DIS3$^{G766R/+}$ and WT genotypes ($n = 8$ DIS3$^{G766R/+}$ $n = 7$ DIS3$^{+/+}$ individuals in immunized and $n = 3$ / genotype for naïve mice; Error bars represent SEM). **e** DZ

(CXCR4$^{high}$ CD86$^{low}$) and LZ (CXCR4$^{low}$ CD86$^{high}$) composition was unaffected ($n = 8$ DIS3$^{G766R/+}$ $n = 7$ DIS3$^{+/+}$ individuals; Error bars represent SEM). **f** MicroC analysis of chromatin architecture at the *Igh* locus in wild-type and *Dis3*$^{G766R/+}$. Splenic B-cells were activated for 3 days in vitro. Displayed is the normalized interaction matrix at the resolution of 2 kb. Ligation frequencies from two replicates per condition were summed before the normalization. **g** Distribution of the normalized MicroC signal representing DNA interactions between the 3′RR super-enhancer and the Eμ intronic enhancer in *Dis3*$^{G766R/+}$ and *Dis3*$^{+/+}$ primary B-cells (Box plots show the median - center line, 25–75th percentiles - box, whiskers to 1.5×IQR, and outliers as individual points; separate for each individual $n = 2$, see box in **f** for the annotation of the interacting genomic intervals). Differences in all pairwise comparisons are not statistically significant. **h** CTCF and **i** Rad21 ChIP-seq signal in the *Igh* locus. All statistical analyses were performed using two-sided, Wilcoxon rank-sum test. Source data are provided as a Source Data file.

## MM DIS3 variants do not affect CSR and genome architecture in B-cells

The nuclear exosome has been demonstrated to be involved in CSR. Both the dysfunction of the nuclear exosome complex[9] and *Dis3* KO[8] abrogate CSR. Therefore, we wanted to analyze the effect of *DIS3*$^{G766R/+}$ mutation on CSR using in vitro-activated primary mouse B-cells. As revealed by cytometric analysis, on average 20% of cells have undergone the switch to the IgG1 heavy chain class at day 3 post-activation in both genotypes, proving that the heterozygous DIS3$^{G766R}$ mutation does not affect CSR (Fig. 2a, Supplementary Fig. S3g). Likewise, the expression of various immunoglobulin heavy- and light-chain classes at days 3 and 5 post-activation was comparable between the two genotypes (Supplementary Fig. S3h). Furthermore, immunization of mice with two anti-COVID-19 vaccines (mRNA and protein-based) shows unaltered production of specific α-Spike IgG1 antibodies, proving unaffected IgM to the isotype IgG CSR as well as SHM (Fig. 2b). The lack of a CSR defect in the mouse model is consistent with data from clinical samples: our analysis of the CoMMpass MM genomic dataset revealed no disruption of CSR in patients with either somatic or germline DIS3 mutations (Fig. 2c).

To look for potentially more subtle differences between DIS3$^{G766R/+}$ and WT mice, we analyzed splenic germinal center (GC) B cells using flow cytometry. Analysis demonstrated a significant expansion of splenic germinal center (GC) B cells (CD95$^+$ GL7$^+$) upon immunization compared with naïve controls (Fig. 2d). However, the magnitude of the GC response was indistinguishable between genotypes (Fig. 2d, Supplementary Fig. S4a–b), and the distribution of B cells within the dark zone (DZ, CXCR4$^{high}$ CD86$^{low}$) and light zone (LZ, CXCR4$^{low}$ CD86$^{high}$) remained comparable (Fig. 2e), resulting in similar DZ:LZ ratios, which on average approached 2 (Supplementary Fig. S4c), indicating that both the quantitative and qualitative features of the GC reaction are preserved in *Dis3*$^{G766R/+}$ mice.

These results together demonstrate both the proper functioning of humoral response in Dis3$^{G766R/+}$ mice in vivo as well as unaffected CSR in MM patients with a DIS3 mutation.

As a next step in *DIS3*$^{G766R/+}$ model characterization, we wanted to assess the structure of the *Igh* locus chromatin architecture, which plays a crucial role in CSR. As we wanted to enhance our power to detect loops, we employed MicroC, a high-throughput micrococcal nuclease (MNase)-based chromosome conformation capture technique[63]. MicroC libraries were generated from activated splenic

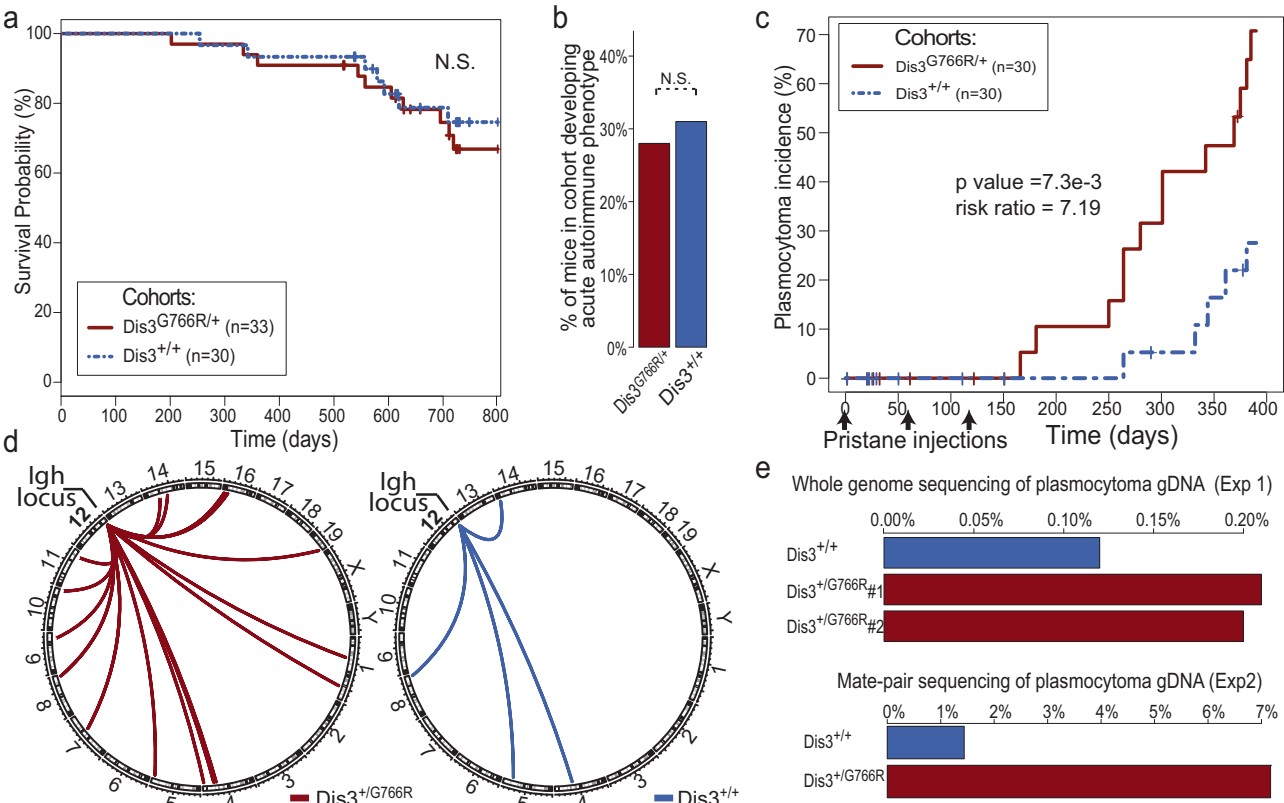

**Fig. 3 | Predisposition to plasmacytoma development in Dis3[G766R/+] knock-in mice is accompanied by frequent *Igh* translocations. a** Kaplan-Meier survival plot of male WT and Dis3[G766R/+] aged mice. **b** Incidence of WT and Dis3[G766R/+] mice developing diffuse alveolar hemorrhage together with a generalized autoimmune phenotype in response to a peritoneal injection of pristane. **c** Cumulative incidence plot showing the frequency of plasmacytoma in Dis3[G766R/+] and WT mice after the pristane injection. Note that the Dis3[G766R] mouse line was constructed on a C57BL/

6 N genetic background, naturally resistant to plasmacytoma development. **d** Circos plot illustrates translocations that involve the *IGH* locus detected in plasmacytoma cells isolated from Dis3[G766R/+] mice (red) and WT mice (green). **e** Fraction of chimeric reads in plasmacytoma samples isolated from Dis3[G766R/+] and WT mice identified using standard whole-genome DNA sequencing (top) and the sequencing of mate-pair libraries (bottom). Source data are provided as a Source Data file.

primary B cells, with two independent replicates per individual per genotype. We sequenced a total of ~1.2 billion (around 600 million reads per condition) MicroC reads. As expected, we observed a plaid pattern of intrachromosomal contacts in both wild-type and mutant B cells. The overall interactomes were similar between conditions; we detected 20,603 and 20,279 loops in the wild-type and heterozygous mutant B cells, respectively (Supplementary Data S1 and S2). Our analysis confirmed the presence of the previously described loops anchored by 3′RR/Eµ and Sγ1/Sµ, in *Dis3[G766R/+]* and *Dis3[+/+]* (Fig. 2f–i). Loop anchors intersected ChIP-seq peaks of CTCF and Rad21[64] (Fig. 2h, i). Moreover, the normalized MicroC signal representing DNA interactions between the 3′RR and the Eµ (blue rectangle, Fig. 2f) did not show statistically significant differences between *Dis3[G766R/+]* and Dis3[+/+] B-cells (Fig. 2g). These results contrast with the previously reported observation in *Dis3* KO B-cells, which showed abolished interactions[8].

Taken together, we demonstrate that Dis3[G766R/+] B cells, despite exhibiting a mild phenotype primarily related to non-coding transcription, display no overt changes in physiology. This includes maintaining exosome function, which is critical for germinal center B-cell differentiation, antibody maturation, CSR, SHM, and the preservation of the chromatin architecture of the Igh locus.

**MM-specific *Dis3[G766R]* mutation drives murine plasmacytoma**
To check the potential general predisposition of *Dis3[G766R/+]* mice to develop cancer, 33 individuals of *Dis3[G766R/+]* and 30 WT were aged under standard conditions for over two years. We observed no significant effect of the MM-associated *Dis3[G766R/+]* allele on survival

(Fig. 3a), nor any signs of accelerated cancer development. To test for an increased spontaneous transformation, we have also cultured MEF cells of both genotypes according to the 3T3 protocol, which models early cancer development and genetic stability[65]. *Dis3[G766R/+]* MEFs did not show an increased frequency of spontaneous immortalization (Supplementary Fig. S5a), arguing against a generalized tendency toward cancerous transformation in cells with MM-associated DIS3 point mutations.

Since *DIS3* mutations are specific to MM and are not commonly found in other cancers, we decided to investigate the development and progression of malignancies of B-cell lineage. We employed a well-known in vivo model using pristane, an adjuvant that induces chronic inflammation, injection into the peritoneal cavity[24,27], which leads to the development of plasmacytoma in susceptible mouse strains[66–68]. Both *Dis3[G766R/+]* and WT C57BL/6 mice exhibited similar immune responses to the pristane injection, indicated by the incidence of diffuse alveolar hemorrhage (Fig. 3b). Despite the natural resistance of the C57BL/6 strain to plasmacytoma, the Dis3[G766R/+] mice developed the disease, with a rate significantly higher than WT controls (risk ratio of 7.19; *p* = 0.0073; Fig. 3c). Plasmacytoma cases displayed characteristic symptoms with atypical plasma cells in an ascites fluid (Supplementary Fig. S5b–d). Notably, *Dis3[G766R/+]* mice developed the condition not only more frequently and rapidly but also accumulated more ascites fluid than WT (Supplementary Fig. S5e–h).

As plasmacytoma is known to depend on aberrant AID-dependent translocations during CSR[24,27], we wanted to investigate the frequency of translocations in Dis3[G766R/+] and Dis3[+/+] mice. DNA was isolated from

the plasmacytomas, and deep sequencing was performed. The analysis of mate-pair and whole-genome sequencing data revealed that plasmacytomas from $Dis3^{G766R/+}$ mice had significantly higher rates of chromosomal translocations involving $Igh$ locus (Fig. 3d, Supplementary Data S3- S6). In addition, more chimeric reads originating from two different genomic locations were identified in $Dis3^{G766R/+}$ (Fig. 3e), indicating increased genome structural instability. Because plasmacytomas are very rare in C57BL/6 mice and largely dependent on the Dis3$^{G766R/+}$ genotype, we additionally backcrossed Dis3$^{G766R/+}$ mice to the plasmacytoma-prone BALB/c background, where plasmacytomas develop more frequently, providing sufficient WT control material for comparison. Notably, the difference in translocation-indicating reads frequency between Dis3$^{G766R/+}$ and WT genotypes was observed in both mouse strains (Fig. 3e).

These results imply that the oncogenic effect of the $Dis3^{G766R/+}$ variant is exerted through the increased mutator effect driving translocations, particularly from the $Igh$ locus, which agrees with the prevalence of $DIS3$ mutation in non-HRD MM.

## Dis3-dependent translocations in mouse plasmacytomas have AID target characteristics

Considering that transient transcription and its degradation at the $Igh$ switch locus is essential for AID-dependent recombination[7,9,17], we aimed to characterize translocations in $Dis3^{G766R/+}$ plasmacytomas.

A meta-analysis of genomic regions surrounding chromosomal breaks detected in plasmacytomas revealed striking enrichment of AID occupancy determined previously by chromatin immunoprecipitation (ChIP-seq) (Fig. 4a). Further analysis showed that both sites of genomic structural variants (SV) and regions of AID activity are GC-rich (Fig. 4b), enriched with AID-specific motifs (Fig. 4c), and are in close proximity to regions predicted to form G-quadruplexes (Fig. 4d). RNA-seq read coverage over SV regions indicates enrichment of steady-state RNA levels over the random genome (Fig. 4e, f, Supplementary Fig. S6). Most of them showed low-level pervasive transcription known to be predominantly cleared by DIS3. When transcribed (Fig. 4g), regions involved in translocations are known to be prone to RNA polymerase stalling and R-loop formation. Indeed, regions involved in structural chromosomal aberrations in our plasmacytoma samples show a level of DRIP-seq accumulation (Fig. 4h–j, Supplementary Fig. S6a–e) like AID off-target loci (Fig. 4h)[69]. Moreover, $Dis3^{G766R/+}$ B-cells accumulate and display an asymmetric distribution of RNA/DNA hybrids with a 5′ to 3′ slope on both DNA strands (Fig. 4h). Since DIS3$^{G766R}$ protein is less efficiently degrading structured RNA[5,6], the observed asymmetry could reflect the stalling of mutant DIS3 along its processive 3′ to 5′ activity. Analogously, the switch regions in the $Igh$ locus show a roughly 25% increase in $Dis3^{G766R/+}$ DRIP-seq signal (Fig. 4i). This can be explained by the fact that Dis3 mutants, including the MM $Dis3^{G766R}$ variant, are expected to have difficulties processing G-quadruplexes enriched in translocating regions (Fig. 4d) and representing very stable RNA secondary structures. Similarly, the TSS regions of MM driver genes, known as AID targets[69], have a significantly stronger DRIP-seq signal compared to the average of all active genes in B lymphocytes (Supplementary Fig. S6f). Consequently, these regions in plasmacytoma are expected to be susceptible to AID-dependent DNA damage[18,70]. A snapshot of the immunoglobulin heavy chain, constant fragments genomic region (the best studied in the context of AID activity) exemplifies all the correlations mentioned above (Fig. 4k).

We conclude that chromosomal translocations caused by the MM-associated DIS3$^{G766R}$ mutation display AID target characteristics.

## Mutational profiles revealed signatures of promiscuous AID activity and a prevalence of translocations in patients with MM $DIS3$ alleles

Having established that specifically in the B-cell lineage, MM $DIS3$ alleles lead to increased genome instability, our objective was to test this finding against clinical data from the CoMMpass MM study, comparing patients with and without $DIS3$ mutations. Patients were categorized based on the type of $DIS3$ mutation, distinguishing between the recurrent mutations (D479, D488, and R780), other $DIS3$ variants, and WT $DIS3$. These groups consisted of 22 patients with recurrent mutations, 48 patients with other variants, and 993 patients with WT DIS3, respectively.

First, analysis of translocations involving the $IGH$ locus revealed their significantly higher occurrence in patients with $DIS3$ mutations (64.3% of non-recurrent $DIS3$ mutated patients, 64.5 % of recurrent $DIS3$ mutated patients, and 33.2% in the rest of the patients) (Fig. 5a, b, Supplementary Fig. S7a, b), agreeing with the specificity of $DIS3$ mutations for non-HD MM and the role of $DIS3$ in AID-dependent translocations.

Then, to characterize the effect of MM $DIS3$ alleles on point mutations, we analyzed 96 COSMIC (Catalog of Somatic Mutations in Cancer) mutational classes of single-base substitution (SBS) profiles[71], estimating the contribution of each of them (Fig. 5c–g, Supplementary Fig. S7c, d). We identified age-associated, exome-wide SBS5 as the primary mutational signature in patients of all $DIS3$ genotypes (Fig. 5c, Supplementary Fig. S7c), in agreement with the late onset of MM and previous analysis[41,72,73]. Similarly, MM-specific mutations detected in $DIS3$ in CoMMpass study patients and mutations reported in the COSMIC database show a predominance of age-associated SBS5 (Fig. 5d). Despite the exosome complex involvement in SHM, only a small fraction of mutations across the exome could be attributed to mutation signatures associated with AID activity across all the $DIS3$ genotypes (Fig. 5c), specifically SBS84 (reflecting canonical AID according to the most recent classification) and SBS85 (concordant with indirect effects of AID activity)[74].

To be more specific, we next analyzed transcribed PROMPT regions toward which DIS3 is targeted. These indeed displayed distinct somatic mutation patterns compared to the entire exome. PROMPTs featured a significantly higher proportion of substitutions that can be attributed to DNA damage, especially SBS3, associated with the erroneous repair of double-strand DNA breaks in a BRCA-dependent mechanism (Fig. 5e). This, in principle, could be explained by the reported DIS3-dependent increase in genome instability accompanying RNA:DNA hybrid accumulation[61]. However, despite PROMPTs being the most prominent DIS3 substrates, the somatic MM $DIS3$ alleles do not affect their AID-dependent mutagenesis (Fig. 5e). Moreover, the contribution of AID-dependent signatures in all transcribed PROMPT regions was limited (Fig. 5e).

To further narrow the search for the effect of the DIS3 mutation, we analyzed known AID targets, which also include a subset of PROMPTs. There, the contribution of AID-dependent signatures is significant and higher in patients with MM $DIS3$ alleles, especially the recurrent ones (Fig. 5f). Further analysis of genomic regions encompassing known AID-dependent MM driver genes[69] where mutations are under positive selection pressure during the carcinogenesis revealed a very strong overrepresentation of signatures related to both AID-dependent mutagenesis and defects in the repair of DNA lesions typical for CSR and SHM (namely: BER SBS10b, SBS30, and MMR SBS6, SBS26) (Fig. 5f, g). Gratifyingly, patients with MM-specific DIS3 variants exhibit a drastic increase in the contribution of these signatures, especially SBS26 (35% in recurrent mutated DIS3 patients, 23% in patients with non-recurrent DIS3 mutations, and 15% in the rest of the patients; Fig. 5g). These results strongly support the positive effect of MM DIS3 variants on AID-dependent mutagenesis.

All analyses presented above indicate that although MM-associated DIS3 alleles do not alter the mutational profile genome-wide, they enhance AID-related mutagenesis occurring within a very limited time window of naive and memory B-cell activation rounds. Furthermore, these mutations are positively selected in very specific loci during MM development.

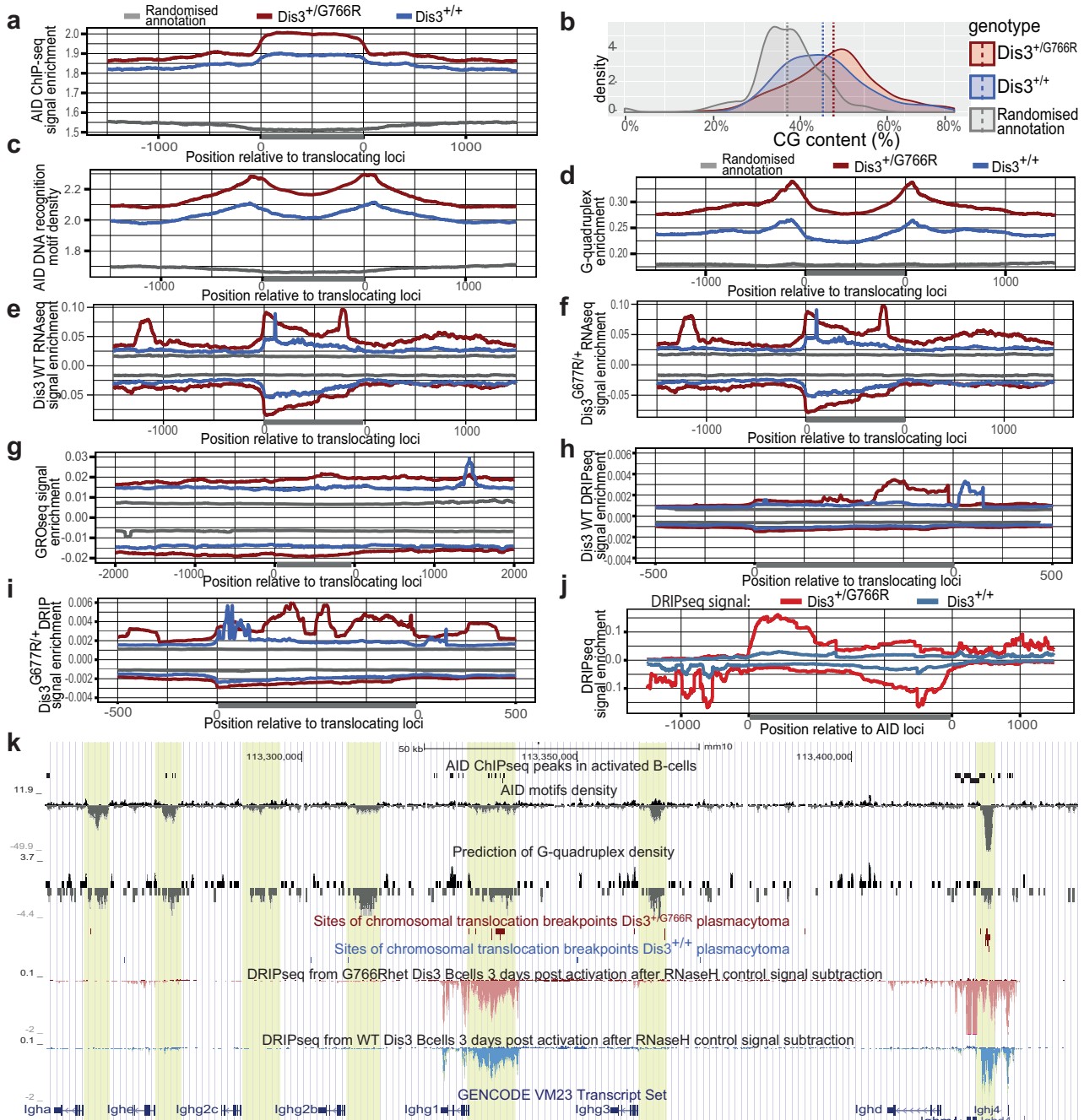

**Fig. 4 | DIS3-dependent translocations in mouse plasmacytomas have AID target characteristics. a–j** Meta-analyses of genomic regions engaged in translocations with features specific for sites of AID activity: AID ChIP-seq signal (**a**), GC-rich regions (**b**), AID-specific DNA recognition sequence motifs (**c**), predicted G-quadruplex sequences (**d**), WT (**e**) and Dis3$^{G766R/+}$ (**f**) RNA-seq of day 3 activated B-cells, native transcription measured by GRO-seq (**g**) and WT (**h**) and Dis3$^{G766R/+}$ (**i**) DRIP-seq signal of day 3 activated B-cells, relative to random genomic distribution. **j** Meta-analyses of DRIP-seq signal from Dis3$^{+/+}$ and Dis3$^{G766R/+}$ over known AID target loci. **k** UCSC genome browser snapshot of a fragment of the *IGH* locus that shows the clustering of translocation sites with regions that coincide with the enrichment of AID-specific recognition sequence motifs, G-quadruplexes, R-loops, active bidirectional transcription, and AID occupancy.

**Mutant DIS3 stalling on chromatin-bound substrates, resulting in increased AID-dependent dsDNA breaks in activated B-cells**

Given the correlation between MM *DIS3* alleles with AID-dependent mutations and translocations in both patients' samples and the murine model, we next wanted to gain insights into the mechanism, which would have to be related to DIS3 association with chromatin.

We found immunoprecipitation of DIS3 from chromatin extremely challenging, possibly because its interaction with chromatin is indirect and transient. Thus, to experimentally verify the direct involvement of MM DIS3 protein variants, we analyzed chromatin

occupancy of the G766R mutant and WT DIS3 protein using the DamID methodology. This approach is based on fusing a protein of interest to DNA adenine methyltransferase (Dam), which modifies adjacent adenines in DNA, providing sufficient sensitivity to detect even weak interactions between DNA and its associated factors[75], which in the case of DIS3, will represent both RNAs engaged in R-loops and RNAs more loosely associated with the chromatin. We modified the established mouse B-cell lymphoma CH12F3-2A cell line, which is capable of cytokine-induced *Igh* CSR. Even before analyzing the sequencing data, we noticed 2.5-fold higher library yields from cells that express

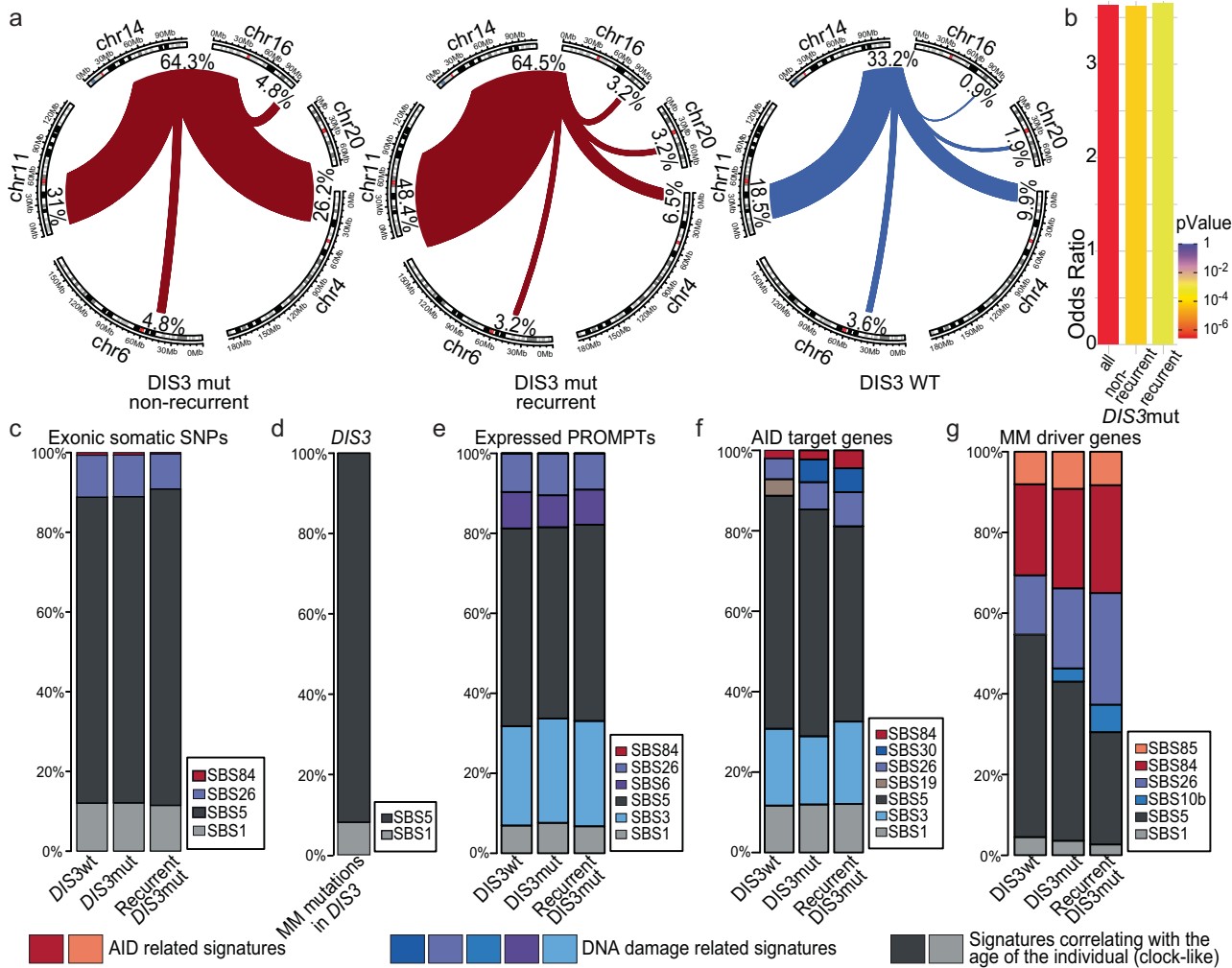

**Fig. 5 | Mutational signatures reveal increases in AID and DNA damage repair lesions on MM driver genes in DIS3 mutant patients. a** Circos graphs depicting the MM driver translocations of the *IGH* locus in WT DIS3 CoMMpass patients *vs.* patients with somatic non-recurrent and recurrent DIS3 MM variants. **b** Bar plot of the enrichment of *IGH* MM-specific translocations in DIS3 mutant patients of different genotypes compared with WT DIS3 patients. (Statistical significance was assessed using a two-sided Fisher's exact test.) **c–g** Contribution of COSMIC mutational signatures in all somatic single-nucleotide variants (SNVs) acquired by CoMMpass patients with WT DIS3, a mutant DIS3, or a recurrent MM DIS3 variant in whole exome sequencing (WES): exome-wide (**c**), in all somatic DIS3 mutations (**d**), expressed PROMPTs (**e**), in genes known to be AID off-targets (**f**) and in MM driver genes (**g**).

DIS3$^{G766R}$ compared with WT DIS3 (Fig. 6a). Such an increase can be explained by a higher chromatin occupancy of the mutated protein and agrees with the previously observed stalling of DIS3 mutant on RNA substrates[6].

DIS3 has a very broad pattern of association with chromatin. However, PROMPT regions, which give rise to the most potent DIS3 substrates (Fig. 6b), and sites of translocations mapped in plasmacytomas show significant enrichment for DIS3 of both genotypes, coinciding with native transcription (Fig. 6b, c). Notably, known AID off-target genes are among the most highly DIS3-occupied loci in the genome (Fig. 6b). Moreover, DIS3 occupies the vicinity of switch regions in the *Igh* locus (Supplementary Fig. S7e), agreeing with the role of DIS3 in CSR. The data presented above indicate that DIS3 acts together with AID to drive translocations. To support cooperation with an orthogonal methodology, we have analyzed interactions between DIS3 and AID using Proximity ligation assay (PLA) (Fig. 6d–e). Indeed, in CH12F3-2A cells, we see PLA-positive spots, the number of which is higher upon in vitro activation (Fig. 6e and Supplementary Fig. S8a).

To further explore causality, we examined DNA damage accompanying B-cell activation in vitro using γ-H2AX, an established marker of dsDNA breaks[76]. Naïve B-cells present a low level of DNA damage,

which, however, increased significantly in cells 3 days post-activation (Fig. 6f, g). In contrast to naïve, activated B-cells demonstrated a moderate, yet statistically significant, increase of DNA damage in *Dis3$^{G766R/+}$* compared to *Dis3$^{+/+}$* (Fig. 6f, g), further supporting the role of MM *DIS3* alleles in B-cell-specific carcinogenesis. To directly test whether the observed increase in DNA damage depends on AID activity, we employed the CH12F3-2A cell line in both AID-proficient (AID$^{+/+}$) and AID-deficient (AID$^{-/-}$) backgrounds. DIS3$^{WT}$ or DIS3$^{G766R}$ were expressed via lentiviral transduction, followed by γ-H2AX dsDNA breaks measurements (Fig. 6h, Supplementary Fig. S8b). Consistent with our primary B-cell data, upon CH12F3-2A cells activation, DIS3G766R expression led to a statistically significant increase in DNA damage compared to DIS3 WT in AID$^{+/+}$ cells (Fig. 6h). Importantly, this effect was strictly AID-dependent, as γ-H2AX levels were highly elevated in DIS3$^{G766R}$ expressing AID$^{+/+}$ cells but not in their AID$^{-/-}$ counterparts. This demonstrates that AID activity is required for the accumulation of DNA damage in the presence of mutant DIS3.

Finally, as AID preferentially targets the coding DNA strand during transcription[77], and displacement of nascent RNA by the nuclear exosome is crucial for AID activity on the template strand[9], we analyzed cytidine somatic mutations in plasmacytoma cells from *Dis3$^{+/+}$* and

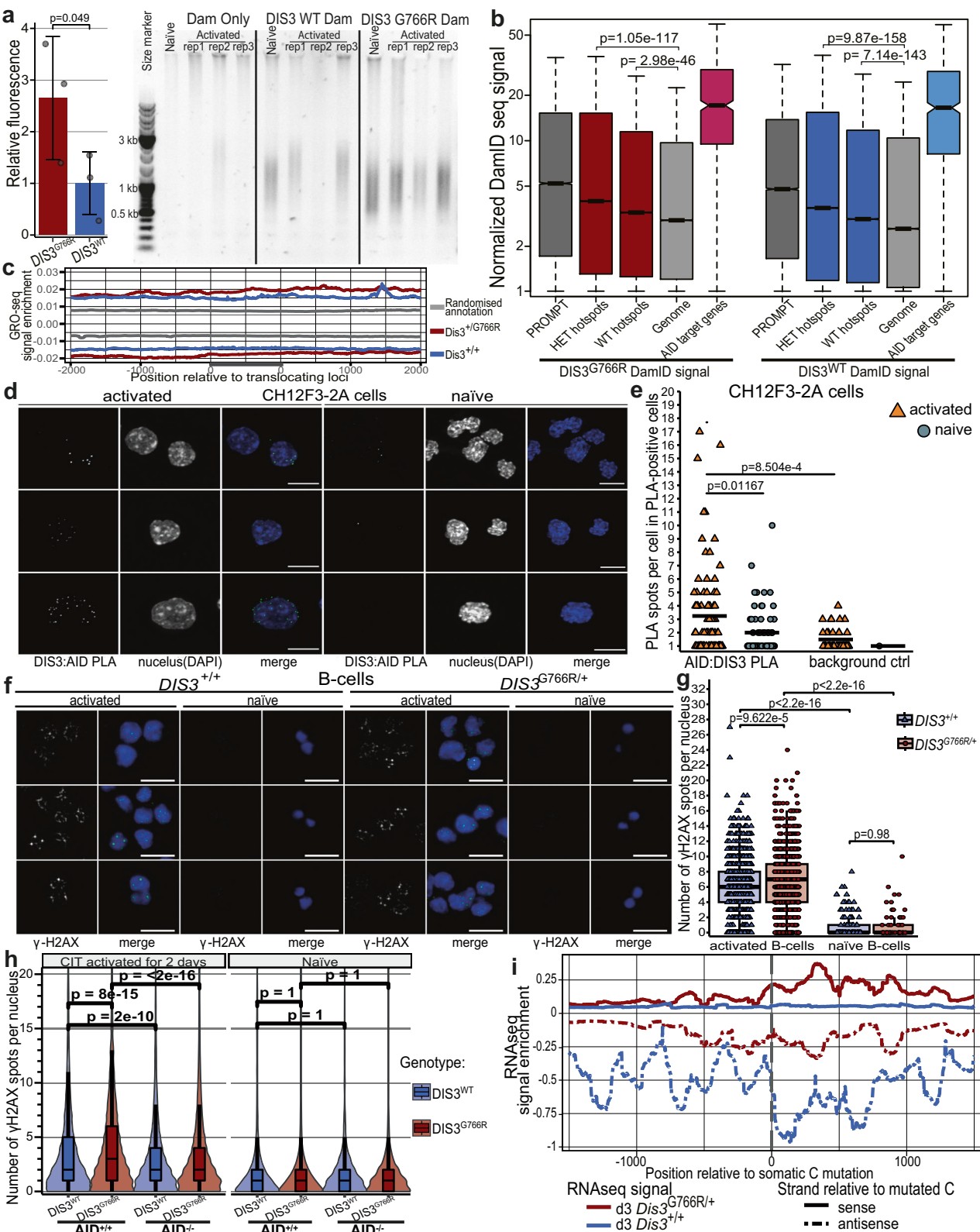

*Dis3*$^{G766R/+}$ mutant mice, comparing their orientation to strand-specific RNA-seq data from B-cells at various activation stages. In DIS3 WT cells, mutations showed a clear strand bias consistent with canonical AID activity (Fig. 6i, Supplementary Data S7-S8). This asymmetry was notably reduced in *Dis3*$^{G766R/+}$ cells, indicating that the DIS3 mutation disrupts AID strand specificity and suggesting that mutant DIS3 stalling increases AID access to the template strand. These findings strongly

support the model in which mutant DIS3 stalling increases AID access to the template strand, resulting in a more symmetrical mutation pattern. When processed by DNA repair pathways, such symmetry increases the likelihood of staggered double-strand breaks—an essential step for both CSR and oncogenic chromosomal translocations[9,78]. What is more, nucleotide-resolution mapping of breakpoints revealed microhomology-mediated repair in both

**Fig. 6 | Mutant DIS3 stalling on chromatin-bound substrates, resulting in increased AID-dependent DNA damage in activated B-cells. a** Mean intensities of DamID libraries obtained from activated CH12F3-2A cells that express WT and DIS3[G766R]. (One-sided unpaired homoscedastic t-test; error bars = SD). **b** DIS3 DamID-seq in activated CH12F3-2A cells shows signal enrichment compared with the rest of the genome over regions that translocate in plasmacytomas and represent known DIS3 substrates, such as PROMPTs (*n* = 3 / genotype; Mood's median test). **c** GRO-seq shows enrichment of native transcription at the same genomic sites. **d** Representative confocal microscopy images demonstrating DIS3 and AID interaction in CH12F3-2A cells detected by in situ PLA (Scale bars = 10μm. Representative images from one high-throughput microscopy experiment are shown; quantification (adjacent panel) is derived from the same experiment. All required controls are present in the Supplementary Fig. S8a). **e** Quantification of the number of DIS3 and AID interaction sites (PLA spots) in CH12F3-2A. Analysis was restricted to cells with at least one PLA spot identified (PLA-positive cells). The black horizontal lines represent means. (Statistics: two-sided, Wilcoxon rank-sum test; *n* = 1076 nuclei in activated and *n* = 209 nuclei in naïve Ch12F3-2A cells) **f** Representative images demonstrating γ-H2AX staining in naïve and day 3 activated B-cells (Scale bars = 10 μm. green, merged with DAPI in blue). **g** Quantification of number of γ-H2AX immunofluorescent spots. (Statistics: two-sided, Wilcoxon rank-sum test; n = 2418 nuclei in activated and *n* = 209 nuclei in naïve). **h** Quantification of high-throughput confocal imaging of DNA double-strand breaks, defined as the number of γ-H2AX foci per nucleus, in naïve and activated CH12F3-2A cells showed that DIS3[G766R] expression led to elevated γ-H2AX levels in an AID-dependent manner. (Statistics: two-sided, Wilcoxon rank-sum test, Bonferroni-corrected, AID[+/+]: *n* = 3959 DIS3[WT] *n* = 5381 DIS3[G766R]; AID[-/-]: *n* = 13087 DIS3[WT] *n* = 12543 DIS3[G766R] activated CH12F3-2A cells). **i** Directional meta-analyses of RNA-seq signals from day 3 activated B cells over somatically mutated cytidines in plasmacytoma samples of DIS3[G766R/+] mice show decreased specificity of AID mutagenic activity for the coding strand compared to DIS3[+/+] counterparts. All box plots show the median (center line), 25th–75th percentiles (box), whiskers to 1.5×IQR.

genotypes, consistent with AID-dependent B-cell neoplasms and MM[79,80], with *Dis3*[G766R/+] cells (Supplementary Fig. S8c) exhibiting modestly longer microhomologies (7 bp vs. 5 bp in WT), suggestive of increased reliance on alternative end joining.

All these results imply that the oncogenic effect of *DIS3* mutations is exerted through the direct impact of the mutant protein in cooperation with AID on DNA integrity, promoting staggered double-strand breaks, fostering oncogenic chromosomal rearrangements.

### *Dis3*[G766R] allele does not affect the chromatin architecture but hijacks it for joining enhancers with proto-oncogenes

DIS3 and the RNA exosome complex were postulated to be essential for suppressing eRNAs from super-enhancers[17] and PROMPTs at active TSSs[81]. We therefore tested whether translocations in plasmacytomas correlate with these regulatory regions. Using publicly available ChIP-seq data from B-cells 3 days post-activation[64], we defined active enhancers and promoters as regions significantly enriched for H3K4me1 and H3K4me3, respectively (FDR < 0.01), overlapping with H3K27ac peaks. Translocating loci were localized in the vicinity to either enhancer-specific H3K4me1 enrichments or active promoter-specific H3K4me3 ChIP-seq peaks with a median distance of 33 or 25 kb for loci detected in WT and *Dis3*[G766R/+] plasmacytomas, respectively (Fig. 7a and Supplementary Fig. S9a–c). The annotated chromosomal breaks also exhibited a non-random distribution with respect to their proximity to p300 ChIP-seq peaks (Fig. 7b and Supplementary Fig. S9c), a marker of active enhancers as well as super-enhancers (Supplementary Fig. S9b).

To further investigate the relationship between chromatin architecture and translocations, we studied TADs by analyzing positions of CTCF and Rad21 ChIP-seq peaks identified in B-cells 3 days post-activation[64] in relation to the translocations found by us in plasmacytomas. Indeed, translocating loci were localized significantly closer to CTCF and Rad21 ChIP-seq peaks than expected from a random distribution in the genome (Fig. 7c,d, and Supplementary Fig. S9d). We detected a median distance of translocating loci from the CTCF peak to be 10.1 kb or 8.6 kb, and from the Rad21 peak to be 10 kb or 8.5 kb, in WT and *Dis3*[G766R/+] plasmacytomas, respectively.

Since in contrast to *Dis3* ablation, the *Dis3*[G766R] allele does not interfere with the chromatin architecture of the *Igh* locus, we closely examined MicroC data for translocations originating in the *Igh* locus. Translocations identified in the plasmacytomas are in regions with strong interaction, indicating that they are in close physical proximity within the three-dimensional nuclear space (Fig. 7e and Supplementary Fig. S10). Similarly to the *Igh* locus, these interactions do not depend on the Dis3 variant (Fig. 7e and Supplementary Fig. S10).

In sum, we demonstrated that although the Dis3 mutant does not alter chromatin architecture, it increases the frequency of translocations between transcriptionally active genomic regions during B-cell activation. This leads to a permanent joining of active promoters and enhancers in developing B cells.

## Discussion

Genes involved in fundamental gene expression pathways can acquire specialized functions in B-cells due to the highly complex processes of generation, diversification, and ultimately, the production of massive quantities of antibodies. Notably, two genes involved in RNA metabolism-DIS3[4] and TENT5C (FAM46C)[82]-play critical roles in the humoral immune response. At the same time, mutations in these genes are specifically associated with MM[1–3], affecting almost 30% of patients. TENT5C is an MM-specific oncosuppressor[2,3,82]. In normal B-cells, TENT5C enhances the expression of immunoglobulins by stabilizing their mRNA through cytoplasmic polyadenylation, which at the same time increases the protein load into the endoplasmic reticulum[83]. Disruption of TENT5C reduces ER stress and promotes proliferation in antibody-producing malignant plasma cells.

Our present study provides a mechanism by which DIS3 drives oncogenic transformation in B cells. Contrary to previous suggestions, dysregulation of the transcriptome is not the cause of the neoplastic transformation. MM *DIS3* alleles also do not lead to ubiquitous genome destabilization. Instead, MM-associated *DIS3* mutations lead to a gain-of-function, which introduces B-cell-specific destabilization of the genome, enabling the acquisition of genetic lesions and giving rise to oncogenic genome rearrangements, such as *IGH* translocations, known to be the primary drivers of MM (Fig. 7f–i).

DIS3 is a primary nuclear exoribonuclease that acts from the RNA 3' end. It targets many types of RNA species, mainly pervasive transcription products that are rapidly cleared. In a mouse model with relatively mild heterozygous Dis3[G766R] mutation, the accumulation of the most potent DIS3 substrates is focal, localized mainly to unannotated, non-coding regions of the genome, and shows no detectable changes in the protein-coding transcriptome as well as the proteome. This can be attributed to the fact that DIS3[G766R] partially retains its activity, showing a significant inhibition over selected substrates, specifically structured RNA[5]. DIS3[G766R] stalling on structured RNA substrates results in increased affinity of the mutant DIS3 to chromatin. In B cells, DIS3 is involved in SHM and CSR, processes that require the activity of AID. Chromatin-stalled DIS3 variants associated with MM facilitate the recruitment of AID, leading to increased and more promiscuous AID activity beyond its usual targets. This heightened activity results in a greater frequency of microhomology-mediated translocations, contributing to genomic instability and potentially driving oncogenic events (Fig. 7g–i). In a mouse model, the heterozygous *Dis3*[G766R/+] mutation increases the rate of plasmacytoma development, accompanied by a significantly higher level of structural genomic variants.

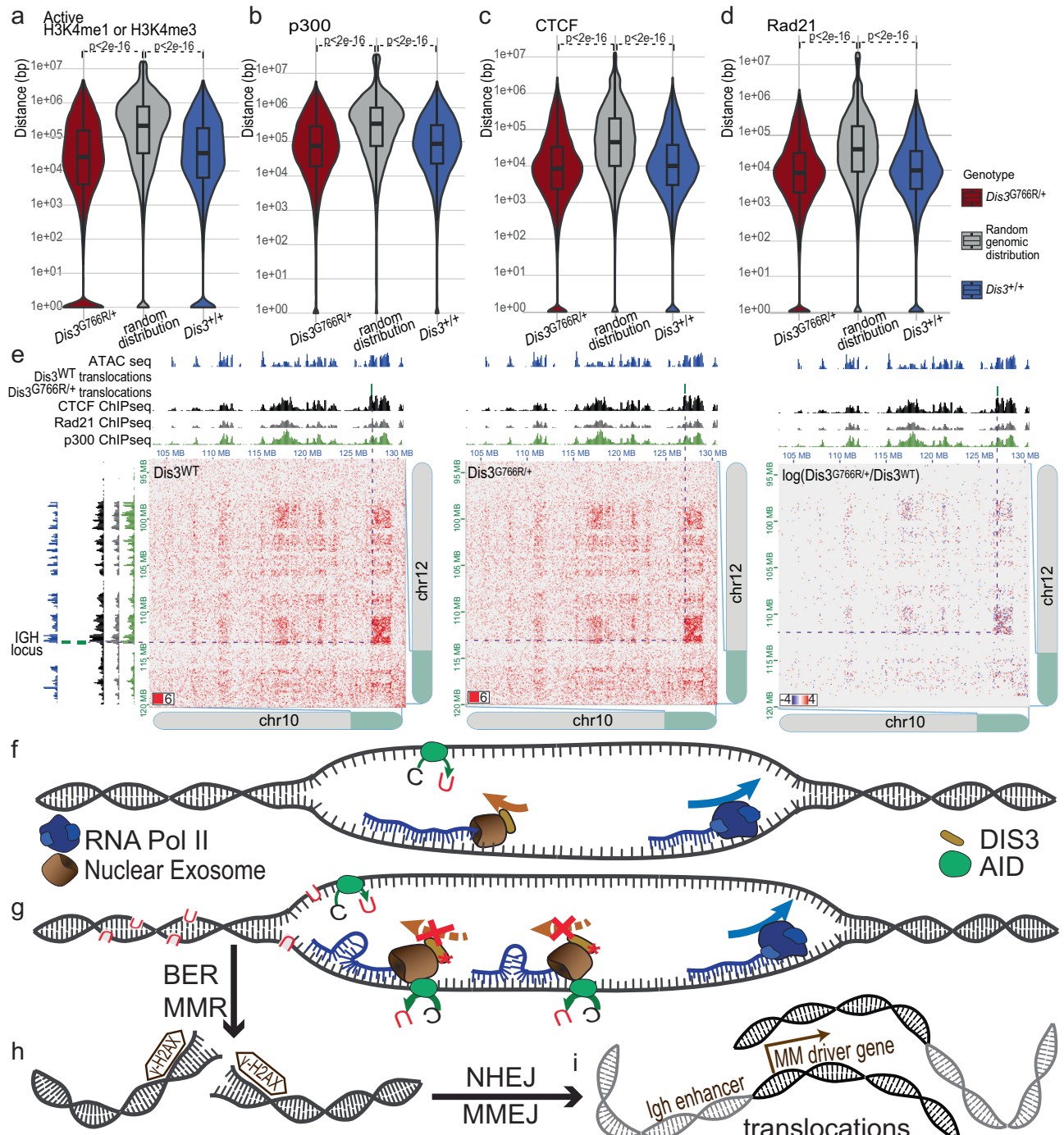

**Fig. 7 | Model of the involvement of MM-specific DIS3 variants in MM onco-genesis through enhanced AID recruitment to ssDNA. a–d** Distributions of distances of regions translocating in murine plasmacytomas to the closest ChIP-seq peaks of: H3K4me1 or H3K4me3, both overlapping with H3K27ac ($n = 55803$ H3K4me1peaks; $n = 25353$ H3K4me3 peaks) (**a**), p300 ($n = 20159$ peaks) (**b**), CTCF ($n = 81036$ peaks) (**c**) and Rad21 ($n = 85771$ peaks) (**d**). (Box plots show the median (center line), 25th–75th percentiles (box), whiskers to 1.5×IQR. Two-sided, Wilcoxon rank-sum test, Bonferroni-corrected). **e** Representative example of an intrachromosomal interaction accompanying Igh-derived translocation t(10,12) in murine-induced plasmacytoma. **f** AID is recruited to DNA during class switch recombination (CSR) through transcription-generated single-stranded DNA regions. These regions serve as substrates for AID, initiating CSR and ultimately

producing antibodies with altered functions. The nuclear exosome with WT DIS3 recruits AID to transcribed sequences, but due to the transient AID activity outside the *Igh* locus, such events are rare. **g** MM DIS3 variants increase AID activity by enhancing the affinity of the nuclear RNA exosome, resulting in increased occupancy/stalling. This results in the accumulation of uracil bases in off-target regions. **h** DNA damage repair mechanisms, including base excision repair (BER) and mismatch repair (MMR) involving uracil glycosylase and AP-endonucleases, process AID introduced uracils, inducing double-strand breaks (DSBs). **i** Non-homologous end joining (NHEJ) and microhomology-mediated end joining (MMEJ) rejoin the free DNA ends, but excessive DSBs can lead to unrestrained genome rearrangements. Source data are provided as a Source Data file.

In physiological conditions, activation-dependent interactions between DIS3 and AID in the B-cell lineage, are confined to discrete sites, comparable to the numbers of DNA damage foci measured by γ-H2AX B-cell activation in vitro. An increased number of γ-H2AX foci in activated CH12F3-2A DIS3[G766R] expressing cells not only confirmed this phenotype but also demonstrated that the dsDNA breaks were AID-dependent, underscoring that AID activity is required for this effect. The statistically significant but moderate increase of DNA damage in Dis3[G766R/+] cells is in contrast to Dis3 depletions, which dramatically affect both cell physiology and genome integrity[61]. Moreover, MM DIS3 mutations display no CSR impairments both in mouse models and in MM samples. The chromatin architecture of the Igh TAD, as well as the rest of the genome, does not show statistically significant changes. Both contrast with Dis3 inactivation, which leads to the rearrangement of tertiary chromatin architecture and inhibits CSR in murine B-cells[8].

A significant effect of the DIS3[G766R] mutation is the disruption of the canonical strand bias in AID-induced somatic hypermutation, resulting in comparable cytosine mutations on both DNA strands. This loss of asymmetry likely increases closely spaced lesions on opposing strands, promoting staggered double-strand breaks and chromosomal translocations. AID activity is normally tightly regulated to prevent excessive off-target mutagenesis[12,70,84]. In DIS3-mutant cells, impaired degradation of structured RNA[5] leads to the accumulation of chromatin-associated transcripts, including G-quadruplexes that can directly recruit AID to nearby ssDNA through its RNA-binding assistant patch. In parallel, persistent R-loops, stabilization of secondary DNA structures such as G-quadruplexes and collapsed R-loops further expose single-stranded DNA substrates with high affinity for AID[21]. The data provided by this study are in line with previous knowledge, indicating that activated AID interacts with replication protein A[85], which anchors it to ssDNA during somatic hypermutation and CSR, while the ensuing DNA damage response activates ATM and PKA, driving AID S38 phosphorylation. Thus, the DIS3-dependent AID recruitment is further enhanced by all these interactions, enacting a positive feedback loop[70]. Collectively, all these effects enhance AID accessibility and persistence on chromatin, promoting off-target activity, genomic instability, and error-prone mismatch mediated end joining (MMEJ) in DIS3 mutant cells.

The nuclear exosome is known to be involved in super-enhancer regulation by controlling levels of eRNA[17]. Interestingly, AID has been shown to promote off-target mutations and genomic rearrangements at super-enhancers, which contribute to oncogene activation and tumor progression[29]. Our data show that translocations detected in murine plasmacytoma predominantly originate in the proximity of enhancer regions as well as active promoters. Translocations originating from the Igh locus in plasmacytoma correlate with inter-chromosomal interactions identified through MicroC data. This aligns with existing evidence that the Igh locus participates in tertiary chromatin structures, interacting in a developmentally regulated manner with specific B cell lineage genes, such as Pax5, Ebf1, Aff3, and Foxp1, many of which overlap with loci implicated in translocations in early B cell malignancies[86]. MM-associated DIS3 mutations may exploit these pre-existing chromosomal interactions, enhancing the likelihood of aberrant translocations. Given the nuclear exosome's role in RNA surveillance within regulatory elements of active gene networks, such mutations can hijack existing chromosomal tertiary structures. In this way, MM-associated DIS3 mutations, even if they are biochemically subtle and have limited transcriptomic effects, can have significant consequences for oncogenesis. A restricted spectrum of DIS3-dependent mutagenesis in MM is supported by analysis of mutational signatures in MM samples, which show that DIS3 mutations are associated with AID-related lesions in MM driver genes, not genome-wide, and occur only in a short window of opportunity during B-cell activation.

In summary, our study shows that mutations in the ubiquitously expressed RNA-degrading enzyme DIS3 enhance its chromatin affinity and AID-dependent mutator activity, driving cancer specifically in the B-cell lineage. It clarifies the role of DIS3 mutations in MM development and explains the specificity of mutant DIS3-induced genetic lesions for MM or, in the case of the mouse model, plasmacytoma.

## Methods

All procedures were approved by the First Local Ethical Committee in Warsaw affiliated at the University of Warsaw, Faculty of Biology (approval numbers WAW/092/2016, WAW/177/2016, WAW/642/2018). Housing in animal facilities was performed in conformity with local and European Commission regulations under the control of veterinarians and with the assistance of trained technical personnel.

### CH12F3-2A cell line

The mouse CH12F3-2A B-cell lymphoma cell line (RCB Cat# RCB2809, RRID:CVCL_E068) was acquired from the Ricken BRC cell bank. Cells were cultured in RPMI 1640 ATCC's modified medium (Invitrogen; A1049101) supplemented with 10% FBS (Invitrogen; 10270-106), 5% NCTC-109 medium (Gibco A/Thermo; 21340039), 50 μM 2-mercaptoethanol, penicillin/streptomycin (Sigma; P4083) at density between 0.5-1 ×106 cells/ml. Media were changed every 3 days.

To obtain lentiviral constructs for the DamID experiment, mouse WT and G766R DIS3 coding sequences were fused to the Dam methylase coding sequence. WT DIS3 cDNA was amplified and cloned into pLgw V5-EcoDam lentiviral constructs (Addgene; 59210) using SLIC. The G766R DIS3 construct was obtained by site-directed mutagenesis.

For the negative control construct, mEGFP was amplified using PCR and cloned into pLgw V5-EcoDam lentiviral constructs (Addgene; 59210) using SLIC. Lentiviral transductions of CH12F3-2A cells were carried out exactly as described previously. Transduced cells were selected using Zeocin at a concentration of 250 μg/ml for 14 days and subsequently, cell cultures were expanded and preserved in 50% FBS, 40% RPMI medium and 10% DMSO.

The mouse CH12F3-2A AID$^{-/-}$ B-cell lymphoma cell line (Kerafast, Cat# ESP013, RRID:CVCL_GZ21) was obtained from Kerafast and cultured under the same conditions as the AID wild-type (WT) CH12F3-2A cells. Both CH12F3-2A WT and AID$^{-/-}$ cells were stably transduced with pLVX-SFFV lentiviral constructs expressing DIS3$^{WT}$, DIS3$^{G766R}$, or mCherry (control). Briefly, lentiviral particles were produced in HEK293T cells by transient transfection using the calcium phosphate precipitation method. Cells were seeded one day prior to transfection to achieve ~70% confluence at the time of transfection. A DNA mixture containing the packaging plasmids psPAX2 (8.6 μg) and pMD2.G (5.5 μg) together with the lentiviral expression construct (8.6 μg, encoding either mCherry, DIS3 wild-type, DIS3$^{G766R}$) was prepared in 55 μL of 2 M CaCl$_2$ and sterile water (final volume 430 μL), then added dropwise under vortexing to an equal volume of 2× HEPES-buffered saline (HBS) to form a white precipitate. The mixture was immediately added to HEK293T cultures. After 16–18 h, the medium was replaced with fresh DMEM containing 10% FBS. Viral supernatants were collected 48 h post-transfection, cleared by centrifugation (800 × g, 5 min) and filtration through a 0.45 μm PES filter.

CH12F3-2A and CH12F3-2A AID$^{-/-}$ cells were transduced with viral supernatant in the presence of polybrene (1:1000) by spin infection (1.5 h, 800 × g, 32 °C). Approximately 3 × 10$^6$ cells per condition were used. Following transduction, cells were cultured in RPMI 1640 supplemented with 10% FBS without antibiotics. Transduction efficiency was monitored by mCherry fluorescence using flow cytometry 48 h post-infection.

CH12F3-2A cells were activated for 2 or 3 days by supplementing the standard culture medium with 1 ng/ml TGF-β1 (Peprotech, 100-21), IL-4 (Peprotech, 214-14)5 ng/ml, 1 μg/ml αCD40 (Invitrogen, 1C10,

Functional Grade, 16-0401-82). Primer sequences are provided in Supplementary Table S1.

## DamID-seq protocol

The DIS3 DamID-seq procedure and library preparations were performed as described previously[87]. In short: DNA from CH12F3-2A cells expressing DIS3 DamID constructs or not, as the control, were isolated using a Bioline Isolate II genomic DNA kit (BIO-52067). First, 0.5 µg of DNA was digested with DpnI (NEB; R0176) in 10 µl volume at 37 °C for 16 h. Next, DpnI was heat-inactivated, and the DpnI adapter was ligated in 20 µl volume with 2.5 units of T4 DNA ligase (NEB; M0202). This was followed by DpnII digestion in a total volume of 50 µl with 10U of DpnII (NEB; R0543). DNA was amplified by PCR with Taq polymerase using the following program: samples were denatured at 95 for 8 min, then amplified for 19 cycles, with: 20" of denaturation at 94 °C, 30" of annealing at 58 °C and 20" elongation 72 °C followed by a 2 min incubation at 72 °C.

PCR products were cleaned up using a Bioline Isolate II PCR and Gel kit according to the manufacturer's instructions (BIO-52060). DNA was repaired with Blunt-end ending 3' overhang using End-It DNA End-Repair kit in 25 µl volume (Epicenter; ER81050), and 3'-A overhang was added with Klenow fragment (NEB; M0212) and dATP. This was followed by purification using CleanPCR beads (CleanNA; CPCR-0005). DNA concentration was measured by Nanodrop and adjusted to 40 ng/µl. Illumina Y-adapter was ligated, and the libraries were PCR amplified using standard protocols and sequenced with an average read number of 40 M. Primer sequences are provided in Supplementary Table S1.

## PLA procedure

In situ Proximity Ligation Assay was performed on activated CH12F3-2A cells and non- activated (naive) cells as a control. Activation was performed by seeding cells at 105 cells/ml in culture medium supplemented with 1 µg/ml anti-CD40 antibody (Invitrogen, MA1-81395), 5 ng/ml IL-4 (Peprotech, 214-14) and 1 ng/ml TGF-β1 (Peprotech, 100-21), and cells were grown for 72 h at 37 °C in 5% CO2. For each experimental condition, 2×106 cells were spun (400 rcf, 4 min), washed with PBS and spun again. Next, cells were fixed with 3,7% formaldehyde in PBS for 25 min at room temperature, washed with PBS twice, permeabilized with 1% Tween-20 (v/v) in PBS for 15 min at room temperature and washed with PBS once. Then, cells were blocked with 3% BSA in PBS for 1 h at room temperature. In order to reduce Fc receptor-mediated binding by antibodies used in PLA procedure, Mouse BD Fc Block (BD Biosciences, 553142) was added to cells in amount of 1 µg/10⁶ cells and incubated according to manufacturer's instructions. Primary anti-AID (1:100; Invitrogen, PA5-18913) and anti-DIS3 antibodies (1:100; Proteintech, 14689-1-AP) were added to cells in blocking solution and incubated overnight at 4 °C. Additionally, three background controls were performed in which cells were incubated only with anti-AID antibody or only with anti-DIS3 antibody, or without primary antibodies. Next, cells were washed with PBS twice and the in-situ PLA was performed with the Duolink In Situ PLA kits (Sigma-Aldrich) according to manufacturer instructions with the exception that the reactions and washings were carried out in 1,5 ml microcentrifuge tubes, and all the incubations were performed in thermoblock. The interaction between AID and DIS3 was detected using Duolink In Situ PLA Probe Anti-Goat MINUS kit (Sigma-Aldrich, DUO92006) and Duolink In Situ PLA Probe Anti-Rabbit PLUS kit (Sigma-Aldrich, DUO92002) against anti-AID antibody and anti-DIS3 antibody, respectively. Fluorescent signal was generated using Duolink In Situ Detection Reagents Red kit (Sigma-Aldrich, DUO92008). Cells were mounted and the nuclei were stained using the Duolink In Situ Mounting Medium with DAPI (Sigma-Aldrich, DUO82040) as follows: after the last wash, cells were spun, all the supernatant was removed carefully, cells were resuspended thoroughly but gently in 12 µl of Mounting Medium, the cell suspension was dripped onto a microscope

slide, covered with a coverslip (Ø 18 mm), and gently pressed down. Edges of the coverslip were sealed with nail polish.

## γH2A.X immunofluorescent staining (B-cells)

DNA double-strand breaks were stained in mouse B cells using γH2A.X as a marker. For each experimental condition, 10⁶ cells were spun (400 rcf, 4 min), washed with PBS and spun again. All the following steps were carried out in suspension in 1,5 ml microcentrifuge tubes. Washed cells were fixed with 3,7% formaldehyde in PBS for 25 min at room temperature, washed with PBS twice, permeabilized with 1% Tween-20 (v/v) in PBS for 15 min at room temperature and washed with PBS once. Then, cells were blocked with 3% BSA in PBS for 1 h at room temperature. Primary anti-γH2A.X (1:400; Abcam, ab2893) was added to cells in blocking solution and incubated overnight at 4 °C. After being washed twice with PBS, cells were incubated with goat anti-rabbit IgG secondary antibody conjugated to Alexa Fluor 555 (1:800; Invitrogen, A-21429) for 1 h at room temperature. Next, cells were washed with PBS once and the nuclei were stained using 2 µg/ml Hoechst 33342 (Invitrogen, H3570) in PBS for 5 min at room temperature. After being washed twice with PBS, cells were spun, all the supernatant was removed carefully, cells were resuspended thoroughly but gently in 12 µL of ProLong Glass Antifade Mountant (P36980, Invitrogen), the cell suspension was dripped onto a microscope slide, covered with a coverslip (Ø 18 mm), and gently pressed down.

## Microscopic analysis

For Fig. 5, cells were imaged using Zeiss LSM800 confocal microscope with 40×/1.3 oil immersion apochromatic objective and GaAsP PMTs detectors. ZEN software (Zeiss, version 2.6) was used for microscope control. Imaging was performed at a controlled temperature of 22 °C. Z-stack images were processed using Fiji/ImageJ software (version 1.53f51). For each Z-stack, the maximum intensity Z-projection was applied and then images of PLA channel were subjected to rolling-ball background subtraction with a radius set to 20 pixels and contrast enhancement with 0.025% saturated pixels.

For Fig. 5 and Fig. EV6, cells were imaged using Olympus IX81-ZDC wide-field fluorescence microscope with 40×/0.95 apochromatic objective and EM-CCD camera (Hamamatsu, C9100-02) equipped with ScanR modular imaging platform (Olympus). ScanR Acquisition software (version 2.2.0.8) was used for microscope control. Imaging was performed at a controlled temperature of 22 °C. Image analysis was performed using ScanR Analysis software (version 3.0.0). Images were subjected to rolling-ball background subtraction with a radius set to 40 and 2 pixels for DAPI and PLA channel, respectively. Next, segmentation was performed on the DAPI channel using the Intensity algorithm. The identified objects were gated based on a scatter plot of the circularity factor relative to the area. The object that comprised the gate was classified as normal, single nucleus. PLA spots in a cell were searched for as sub-objects in the area encompassing a nucleus and a 7-pixel radius outside it, basing on PLA channel and using the Intensity algorithm. Each identified sub-object was assumed to be a PLA spot. Further numerical data processing and statistical analysis were performed using R environment. Cells containing at least one PLA spot were filtered out and for each experimental condition, a total number of spots in a cell and its mean were calculated. Statistical significance was calculated using the Wilcoxon test. All data were checked beforehand for normality using the Shapiro-Wilk test.

For Fig. 5f,g cells were imaged using Olympus IX81 wide-field fluorescence microscope with 60×/1.35 oil immersion apochromatic objective and CCD camera (Hamamatsu, Orca-R2/C10600) equipped with ScanR modular imaging platform (Olympus). ScanR Acquisition software was used for microscope control. Imaging was performed at a controlled temperature of 22⁰C. Image analysis was performed using ScanR Analysis software (version 3.0.0). Images were subjected to

rolling-ball background subtraction with a radius set to 75 and 2 pixels for Hoechst and γH2A.X channel, respectively. Next, segmentation was performed on the Hoechst channel using the Intensity algorithm. The identified objects were gated based on a scatter plot of the circularity factor relative to the area. The object that comprised the gate was classified as normal, single nucleus. γH2A.X foci in a cell were searched for as sub-objects in the area encompassing a nucleus basing on γH2A.X channel and using the Intensity algorithm. Each identified sub-object was assumed to be a DNA double-strand break. Further numerical data processing and statistical analysis were performed using R environment. For each experimental condition a total number of DSB in a cell was calculated. Statistical significance was calculated using the Wilcoxon test.

For Fig. 5f, representative images obtained by the above-mentioned imaging procedure were selected. Images were processed using Fiji/ImageJ software (version 1.53f51). Images of both channels were cropped to provide a more accurate visualization of γH2A.X foci in the cells. Images of γH2A.X channel were subjected to rolling-ball background subtraction with a radius set to 3 pixels and contrast enhancement with 0.01% saturated pixels. Images of Hoechst channel were subjected to rolling-ball background subtraction with a radius set to 40 pixels

### γH2A.X staining and imaging of CH12F3-2A cell lines

DNA double-strand breaks were detected in mouse B cells using γH2A.X (phospho S139) as a marker. Cells ($10^6$ per condition) were washed once with PBS and stained with Live/Dead Fixable Green viability dye (Invitrogen) for 30 min at room temperature, protected from light. Cells were washed twice with PBS containing 1% BSA, fixed in freshly prepared 2% paraformaldehyde (PFA) in PBS for 15 min at room temperature, and washed twice with PBS + 1% BSA. Permeabilization was performed with 0.1% Triton X-100 in PBS for 15 min at room temperature and counterstained with 2 µg/ml Hoechst 33342 (Invitrogen, H3570), followed by two washes with PBS + 1% BSA.

Cells were stained with anti-γH2AX (phospho S139)-PerCP-eFluor 710 (1:200; Invitrogen, 46-9865-42; RRID:AB_2573918) diluted in permeabilization buffer (PBS + 0.1% Triton X-100 + 1% BSA) for 1 h at 4 °C, protected from light. After staining, cells were washed twice with PBS + 1% BSA and resuspended in the same buffer. For imaging, cell suspensions were transferred to Greiner µCLEAR® black poly-D-lysine–coated plates (cat. #781946) and mounted. Images were acquired using the Opera Phenix high-content confocal microscope (PerkinElmer) equipped with a 63× water-immersion objective. Three confocal planes were collected per field.

Image analysis was performed in Harmony software (PerkinElmer). Nuclei were segmented based on Hoechst 33342 signal, and only live cells–negative for the Live/Dead stain–were included in subsequent analysis. γH2A.X foci were automatically detected as discrete spots in the PerCP-eFluor 710 channel.

### Mouse line generation

The constitutive knock-in G766R mice were constructed according to the strategy described in Supplementary Fig. S1b in the Institut clinique de la Souris (ICS) Illkirch, France using homologous recombination technology in mES cells in the C57BL/6 N background.

In brief, embryonic stem (ES) cells were cultured following the standard protocol[88] and then electroporated with the targeting vector. This vector carried a G766R point mutation introduced into exon 17, specifically a G to C substitution at position 2393 of the gene sequence. The mutation site was flanked by 4 and 3.5 kb homology arms (Supplementary Fig. S1B) and accompanied by a floxed CMV promoter-driven Neo-cassette expressing Cre-ER, which was inserted into the intron of the Dis3 gene. Following targeting, ES cells were selected using G418, and clones were genotyped to identify those carrying the Dis3G766R allele. These cells were treated with tamoxifen to remove the cassette, leaving a single loxP site in intron 19. ES cells carrying the Dis3G766R allele were subsequently injected into C57BL/6 N blastocysts, which were then implanted into the uterine horns of pseudopregnant C57BL/6 N mice. Chimeras and their subsequent progeny were mated with C57BL/6 N wild-type mice to generate germline production and expand the colony. The progeny resulting from this cross-breeding produced heterozygous mice Dis3G766R/+ as well as wild-type (WT) controls. Mice are genotyped by PCR using primers: 766seq_F gtctcaaaaacacacggcagagactc; 766seq_R gaggatgtgtgagataag atacaccgg

Dis3 WT allele migrates as a 202 bp band, whereas the mutant G766R allele gives a 281 bp amplicon.

### Mice breeding conditions

Mice were bred in the animal house of Faculty of Biology, University of Warsaw maintained under conventional conditions in a room with controlled temperature (22 °C ± 2 °C) and humidity (55 ± 10%) under a 12 h light/12 h dark cycle in open polypropylene cages filled with wood chip bedding (Rettenmaier). The environment was enriched with nest material and paper tubes. Mice were fed ad libitum with a standard laboratory diet (Labofeed B, Morawski). Animals were closely followed-up by the animal caretakers and researchers, with regular inspection by a veterinarian, according to the standard health and animal welfare procedures of the local animal facility. Mice were euthanized by cervical dislocation in accordance with institutional and ethical committee–approved protocols.

### Mouse embryonic fibroblast primary cell cultures

Primary cell cultures of mouse embryonic fibroblasts (MEF) were isolated from E13.5 embryos as defined by the date of the vaginal plug and confirmed by characteristic features of the embryo morphology. Embryo heads were used for DNA isolation and genotyping. The corpses were gutted and mechanically disintegrated by chopping and passing the tissue fragments suspended in culture media (DMEM supplemented with 10% FBS, 50 µM 2-mercaptoethanol, penicillin/streptomycin) with a syringe through the gauge 19 needles. Disintegrated tissue fragments of a single individual were left to settle in 100 mm petri dishes and form outgrowths. After 3 days, cells were treated with 0.25% Trypsin in 1 mM EDTA for 3 min. Cells were passaged for at least three passages before they were collected or used for further applications. Cells were passaged every 3 days. Cells isolated from three independent litters in pairs, including DIS3$^{+/+}$ and DIS3$^{G766R/+}$ were used for the RNA-seq library preparation.

For the 3T3 protocol, MEF cells from 12 DIS3$^{+/+}$ and 13 DIS3$^{G766R/+}$ littermates were used. Cells passaged two times to disintegrate into a single cell suspension and noted. Cells were plated into a 10-cm cell culture dish at a density of $9 \times 10^5$ cells per dish. Cells were cultured at standard conditions with medium change after 2 days in culture. Three days after plating, cells were trypsinized, counted using a hemocytometer, and replated $9 \times 10^5$ into a 10 cm cell culture dish. The strict passaging regime was repeated for 30 passages or until the immortalization of cells was obtained. Cumulative cell numbers were plotted.

### Pristane-induced mouse plasmacytoma (MPC) model

Six- to ten-week old mice were injected 3 times intraperitoneally, in 2-month intervals, with 0.5 ml of pristane oil (2,6,10,12-tetra-methylpentadecane) (Sigma-Aldrich, Cat. No. P2870) to induce the development of mouse plasmacytoma (MPC), an established murine model of MM[66,67]. The day of the first injection is considered as day 0 of the experiment. Mice were regularly observed for abdominal swelling. The neoplastic nature of the observed ascites was confirmed by paracentesis and ascitic smear stained with Wright-Giemsa stain which had to confirm >10 atipical plasma cells. Animals were closely followed-up by the animal caretakers and researchers, with with particular attention to lupus-like syndrome signs such as alveolar

hemorrhage or fur loss. Regular inspection by a veterinarian was conducted, according to the standard health and animal welfare procedures of the local animal facility. Humane endpoint: After sample collection, mice were returned to the housing facility and monitored for no longer than one month. Mice showing any signs of suffering, significant distress, or significant weight loss (greater than 15% of body weight) were humanely euthanized by cervical dislocation and subsequently sectioned.

Mice for the pristane-induced mouse plasmacytoma experiment were maintained in individually ventilated cages in a room with controlled temperature (22 °C ± 2 °C) and humidity under a 12 h light/12 h dark cycle.

As plasmacytomas in a C57Bl/6 background are a rare event in the WT genotype, to obtain a sufficient amount of material for and better compare plasmacytomas from $Dis3^{G766R/+}$ and WT mice, we have backcrossed $Dis3^{G766R/+}$ mouse line into plasmacytoma prevalent BALB/C background (more than 6 crosses) and used WT and $Dis3^{G766R/+}$ littermates to induce plasmacytomaTo obtain G766R mice on a BALB/C genetic background, $DIS3^{G766R/+}$ mice were backcrossed to BALB/CanNCrlCmd mice for 6 generations.

### Tissue collection and primary cell culture, in vitro B-cell activation, CSR efficiency assay

Splenic naïve B-cells were prepared using the CD43 negative selection commercial kit (Miltenyi Biotec.; 130-090-862). Spleens were pulled from 2-3 terminated sex- and age-matched animals mechanically disintegrated and filtered through a 70 µm cell strainer to prepare a single cell suspension essential for microbead separation. Spleen extracts were depleted from red blood cells using ACK lysis buffer. The magnetic bead, column-based enrichment of naïve B-cells was performed according to the manufacturer's instructions. Cells were resuspended and cultured in RPMI 1640 containing 15% FBS supplemented with 100 nM 2-mercaptoethanol and penicillin/streptomycin (Sigma; P4083).

Ex vivo activation of B lymphocytes for CSR, proliferation studies and RNA sequencing and was achieved by culturing them in RPMI 1640 containing 15% FBS supplemented with 100 nM 2-mercaptoethanol, 20 µg/ml LPS and 20 ng/ml IL-4. CSR efficiency was assessed in B-cells 3 days after activation. In brief, cells were collected and stained for 30 min with Alexa Fluor 555 conjugated goat antibody raised against mouse IgG1 (Invitrogen; A-21127) at 10 µg/ml. The percentage of positive cells that underwent IgG1 CSR was quantified using flow cytometry on an Attune NxT Cytometer (Thermo) or CytoFLEX Flow Cytometer (Beckman Colter) and analyzed using FlowJo (v10.7.1) software.

### CellTrace Proliferation Assay

CellTrace™ Yellow Proliferation kit was used according to the manufacture's instructions. Briefly: Cells were stained by adding 1 µl of CellTrace™ Yellow (Invitrogen, Cat #C34567) stock solution in DMSO per each ml of $10^6$/ml cell suspension in PBS (5 µM working concentration), and subsequently incubated for 40 minutes at room temperature, while being protected from light. Afterward, five times the original staining volume of culture medium, containing FBS, was added to the cells and incubated for 5 minutes. The cells were then pelleted and resuspended in a fresh pre-warmed complete culture medium. 10 minute incubation period was allowed before analysis to enable the acetate hydrolysis of the CellTrace™ reagent. Cells were pipetted in technical triplicates, $3 \times 10^5$ per well, into round bottom 96 well plate in 200 µl growth medium. Cells were activated by supplementing media with 20 µg/ml LPS and 20 ng/ml IL-4 and cultured for 3 days. Naïve B-cells and unstained control cells were parallelly cultured in triplicates. Before cytometric analysis, dead cells were stained with LIVE/DEAD fixable dyes (Invitrogen™) according to manufactures instructions and analyzed on a CytoFLEX Flow Cytometer (Beckman Colter). Data were analyzed using FlowJo (v10.7.1) software using the Proliferation functionality.

### Splenic Germinal Center Phenotyping

Spleens were harvested from naïve and immunized mice (three biological replicates per genotype for naïve animals and eight individuals per genotype at day 14 post-immunization with the Moderna mRNA vaccine). Single-cell suspensions were prepared and analyzed by flow cytometry to quantify and phenotype germinal center (GC) B cells and to determine the light zone (LZ) to dark zone (DZ) ratio.

**Tissue Processing.** Spleens were placed into a 70 µm cell strainer and mechanically dissociated using the plunger of a 5-mL syringe. The strainer was rinsed with 5 mL of cell culture medium (RPMI-1640 supplemented with 10% FBS), and the suspension was transferred to a 15 mL Falcon tube. Cells were pelleted by centrifugation at 800 × g for 3 min at room temperature, the supernatant was discarded, and the pellet was resuspended in 1 mL of ACK lysis buffer to remove erythrocytes. After 5 min incubation at room temperature, 9 mL of culture medium was added to quench the lysis. Cells were centrifuged again (800 × g, 3 min), the supernatant discarded, and the pellet resuspended in 10 mL of culture medium. Cell concentration and viability were determined using a hemocytometer or automated cell counter.

**Flow Cytometry Staining.** For surface staining, $1–2 \times 10^6$ splenocytes were incubated with Live/Dead Fixable Near-IR dye (Invitrogen, L34975; 1:1000 dilution) for 30 min at room temperature, protected from light. Cells were then blocked with anti-mouse CD16/CD32 (Fc block, 1 µg per $10^6$ cells; BD Biosciences) or purified mouse IgG (1–2 µg per $10^6$ cells) in PBS containing 1% BSA for 15 min on ice. Without washing out the blocking reagent, fluorochrome-conjugated antibodies were added in PBS + 1% BSA and incubated for 30 min at 4 °C, protected from light.

The following antibodies were used: anti-CD45R/B220–APC (clone RA3-6B2, BD Biosciences, Cat. #553092, used at 0.5 µL per $10^6$ cells; RRID:AB_398531), anti-GL7– CoraLite® Plus 488 (clone GL-7, Proteintech Cat No. CL488-65261; RRID:AB_3084186; used at 0.8 µg per 10^6 cells), anti-CD95–BUV395 (clone Jo2, BD Biosciences, Cat. #740254; RRID:AB_2739999; at 1 µL per $10^6$ cells), anti-CD184 (CXCR4) Monoclonal Antibody (2B11), Invitrogen eBioscience™, Cat. #14-9991-82; RRID:AB_842770; used at 0.5 µg per $10^6$ cells), and anti-CD86– BD Horizon™ BV510 (clone GL1, BD Biosciences Cat. #563077; RRID:AB_2737991; at 2.5 µL per $10^6$ cells). Samples were acquired on a CytoFLEX LX Flow Cytometer (Beckman Colter) flow cytometer and analyzed using CytExpert (v2.23, Beckman Colter) final visualization and statistical analis were performed in R. Dead cells and doublets were excluded by standard gating. Gating strategy was demonstrated on representative samples in Supplementary Fig. S4A-B.

**Germinal Center Zonal Analysis.** GC B cells were identified as B220$^+$ GL7$^+$ CD95$^+$. Within this population, dark zone (DZ) and light zone (LZ) subsets were defined as follows:

- DZ B cells: CXCR4$^{high}$ CD86$^{low}$
- LZ B cells: CXCR4$^{low}$ CD86$^{high}$

The LZ:DZ ratio was calculated as the frequency of LZ B cells divided by the frequency of DZ B cells within the total GC B-cell gate.

### RNA-seq, Mate-Pair and Paired-End DNA-seq

Total RNA from collected tissues and primary cultures was isolated using a standard TRIzol protocol followed by ribodepletion with a Ribo-Zero Gold rRNA Removal Kit (Human/Mouse/Rat) according to the manufacturer's instructions (Epicenter; RZG1224). Strand-specific libraries were prepared using a TruSeq RNA Sample Preparation Kit (Illumina) and the dUTP method. Three independent replicate sample sets were prepared for each condition. DNA was purified using a SherlockAX kit optimized for small samples (A&A Biotechnology; 095-

25). NEXTERA XT mate-pair and TruSeq DNA libraries were prepared according to manufacturer's instructions (Illumina; FC-131-1024 and FC-121-2001). All of library types were sequenced using Illumina NextSeq or NovaSeq platform in a paired-end mode with different average sequencing depth: MEF RNA-seq: 29 M reads, Mate-pair DNA libraries: 42.7 M reads, standard DNA-seq libraries: 281.5 M reads (NovaSeq). The total RNA extracted from splenic primary B-cell cultures, naïve and in vitro activated for 72 and 96 h has been commercially sequenced by Azenta Life Sciences. They utilized their strand-specific library protocol, achieving an average depth of 23 million 100-nucleotide paired-end reads.

## DRIPc-seq (DNA/RNA hybrid immunoprecipitation followed by cDNA conversion and sequencing)

After 72 h, the culture medium was removed, and cells were washed once with cold PBS and twice with TE buffer (10 mM Tris [pH 8.0] and 1 mM EDTA). Cell lysis was performed directly on the plate by adding 0.5 ml of lysis buffer (TE supplemented with 0.625% SDS and 0.0625 mg/ml 7 Proteinase K), and the suspension was immediately transferred to a fresh 2 ml tube and incubated for 4.5 h at 37 °C with gentle mixing every 15 min. Afterward, genomic DNA was purified by phenol:chloroform:isoamyl alcohol (25:24:1) extraction and ethanol precipitation. Precipitated DNA was washed with 70% ethanol, air-dried, and resuspended in 30 μl of TE buffer. Next, pure gDNA was sheared by sonication using Bioruptor Plus with a water-cooling system (Diagenode) as the following: 3 μg gDNA diluted to 200 μl in TE was placed in 1.5 ml DNA LoBind Tubes (Eppendorf) and subjected to 30 cycles of sonication (30 s ON, 90 s OFF) in "Low" mode. The length of fragments after sonication was evaluated by running DNA samples before and after shearing on a 1.5% agarose gel in 1X TAE (40 mM Tris [pH 8.0], 20 mM acetic acid, and 2 mM EDTA) at 70-80 V. To ensure specificity, nucleic acids are pre-treated with Ribonuclease H (RNase H) (NEB, cat. No. M0297S) to eliminate RNA:DNA hybrids from the mixture. In this process, 4 μg of digested gDNA is incubated with 4 μL of NEB RNase H for 4 h at 37 °C. Following this treatment, proceed with the S9.6 immunoprecipitation step, processing both the control, RNase H-treated samples alongside the experimental samples. Afterward, 4 μg of fragmented gDNA was mixed with 10 μg of S9.6 antibody, diluted to a final volume of 1 ml with binding buffer (20 mM HEPES [pH 7.4], 150 mM NaCl, and 0.05% Triton X-100), and incubated overnight with gentle rotation in a refrigeration cabinet. The next day, input fractions that contained S9.6 were added to 50 μl of M2 anti-FLAG Magnetic Beads (Sigma) or Protein G Dynabeads (Thermo), respectively, and incubated for 2 h. Additionally, a second set of negative controls were prepared by incubating 4 μg of sonicated gDNA with beads without the bait. Following incubation, the unbound fraction was removed, and beads were washed three times with 500 μl of binding buffer for 5 min with gentle rotation in a refrigeration cabinet. Elution was performed by adding 125 μl of elution buffer (50 mM Tris [pH 8], 1 mM EDTA, 0.5% SDS, and 0.64 mg/ml Proteinase K) for 45 min at 55 °C. Two replicates of each sample were pooled and purified using a DNA clean and concentrator kit (Zymo Research) according to the manufacturer's protocol. After elution with 30 μl of elution buffer (Zymo Research), half of the eluate was treated with 2 μl of DNAse I (1 U/μl) in 1X DNase I digestion buffer (Thermo Fisher Scientific) for 1 h at 37 °C. For DNase I inactivation, 1 μl of 0.5 M EDTA was added for 10 min at 65 °C. RNA was purified using the RNA clean and concentrator kit (Zymo Research). The purified RNA underwent NGS library preparation and subsequent sequencing, which was conducted commercially by Azenta Life Sciences using their strand-specific protocol.

## MicroC

MicroC was performed according to[89] with minor modifications. Two replicates of B cells, isolated from WT and Het mice, were activated for 72 h by incubation with LPS and IL4 (20 μg/ml LPS and 20 ng/ml IL-4 final concentration) prior to experiment. Next, cells were collected by trypsinization, washed in PBS and counted. For crosslinking, 300 mM DSG (ThermoScientific) solution in DMSO (100x) was prepared freshly and diluted to 1x in room temperature PBS. Cellular pellet was resuspended in DSG solution and incubated for 35 min on rotating wheel at room temperature. Next 16% methanol-free formaldehyde (Thermo Scientific) was added to a mixture to a final concentration of 1% and incubated for next 10 minutes on rotating wheel at room temperature. To quench the crosslinking reaction, Tris pH 7.5 was added to a final concentration of 0.375 M and incubated on rotating wheel for 5 min at room temperature. Next, cells were centrifuged at 4 °C, washed once with ice-cold PBS and re-counted. Cells were divided into 5 mln cells portions, pelleted, snap frozen and stored at -80 °C until needed.

Before MicroC protocol, MNase titration was performed to specify the required amount of enzyme for sufficient chromatin cleavage. Next, 10 million crosslinked activated B cells from WT and Het mice were resuspended in MB#1 buffer (10 mM Tris pH 7.5, 50 mM NaCl, 5 mM NgCl2, 1 mM CaCl2, 0.1% Igepal, EDTA-free protease inhibitor cocktail (Roche)) and incubated on ice for 20 minutes to isolate nuclei. After one wash with MB#1 buffer, nuclei were resuspended in MB#1 buffer containing MNAse and incubated for 20 min in 37 °C with 1000 rpm rotation in thermoblock to digest chromatin. Reaction was stopped by addition of 4 mM EGTA and 10 minutes incubation at 65 °C. Next, nuclear pellet was washed twice in MB#2 buffer (10 mM Tris pH 7.5, 50 mM NaCl, 10 mM MgCl2, BSA), resuspended in 100 ul Repair End Master Mix (1x NEB 2.1, 2 mM ATP, 5 mM DTT, 50 U PNK), incubated at 37 °C for 15 min with shaking 1000 rpm. Next, 50 U Klenow was added and incubated for next 15 min at 37 °C with shaking 1000 rpm. To biotinylate, Biotin Master Mix (66 nM Biotin-dATP, 66 nM Biotin-dCTP, 66 nM dTTP, 66 nM dGTP, T4 DNA ligase buffer, 33 ug BSA) was added and incubated 45 min 25 °C with shaking 1000 rpm. To quench the reaction EDTA was added to final concentration of 30 mM, incubated at 65 °C for 20 min and the chromatin pellet was washed once with MB#3 buffer (50 mM Tris pH 7.5, 10 mM MgCl2, 50 μg BSA). Next, the chromatin pellet was resuspended in Ligation Master Mix (T4 buffer, 100 μg BSA, 10,000 U T4 DNA ligase), incubated for 2.5 h at 25 °C and centrifuged. The chromatin pellet was resuspended in Exonuclease Master Mix (NEBuffer 1, 1000 U Exonuclease III), incubated for 15 min at 37 °C and centrifuged. Next, the samples were incubated overnight at 65 °C in Reverse crosslinking solution (1% SDS, 200 mM NaCl, 520 ug Proteinase K, 26 ug RNAseA), DNA was purified on 1.0x AmPure beads to enrich for dinucleosome fraction, concentration was measured and aliquot was checked on agarose gel to ensure efficient MNase cleavage.

Prior to library preparation 10–15 ug biotinylated chromatin was pull-down using Dynabeads MyOne Streptavidin T1 for 30 mins incubation at room temperature, washed twice with 1xTWB (5 mM Tris-HCl pH 7.5, 0.5 mM EDTA pH 8.0, 0.1% Tween-20) and once with 10 mM Tris pH 7.5.

Library was prepared using NEBNext® Ultra™ II DNA Library Prep Kit for Illumina® according to manufacturer protocol. Briefly, beads were resuspended in 25 ul water, 3.5 ul End Prep Reaction Buffer and 1.5 ul End Prep Enzyme Mix and incubated in thermocycler for 30 min at 20 °C, followed by 30 mins at 65 °C. Next, 15 ul Ligation Master Mix, 0.5 ul Ligation Enhancer and 2.5 ul unique xGen UDI-UMI Adapter with indexes (IDT) were added and incubated for 30 min at 20 °C. The beads were washed twice with 1xTWB and once with 10 mM Tris pH 7.5. Test PCR using Q5 NEB Mix was run on 10% beads to establish required number of PCR cycles and final PCR was performed accordingly. Eventually, PCR product was purified with two-sided AmPure size selection (0.6x and 1x), measured using Qubit and checked on TapeStation. The libraries were paired-end 150 bp sequenced to around 300 MR each.

## Global proteomics by liquid chromatography (LC)-MS

Cell pellets were subjected to the Sample Preparation by Easy Extraction and Digestion (SPEED) protocol[90]. In brief, cells were solubilized in concentrated TFA and incubated for 10 minutes at RT. Samples were neutralized by adding 2 M Tris-Base buffer using 10×volume of TFA and further incubated at 95 °C for 5 min after adding Tris(2-carboxyethyl)phosphine (TCEP) to a final concentration of 10 mM and 2-Chloroacetamide (CAA) to a final concentration of 40 mM. Protein concentrations were determined by turbidity measurements at 360 nm, adjusted to the same concentration using a sample dilution buffer (2 M TrisBase/TFA 10:1 (v/v)) and then diluted 1:4-1:5 with water. Digestion was carried out overnight at 37 °C using trypsin at a protein/enzyme ratio of 100:1. TFA was added to a final concentration of 2% to stop digestion. The resulting peptides were TMT labeled on-StageTip. TMT-labeled samples were compiled into a single TMT6 sample and concentrated using a SpeedVac concentrator. Peptides in the compiled sample were fractionated (6 fractions) using the bRP fractionation. Prior to LC-MS measurement, the peptide fractions were resuspended in 0.1% TFA, 2% acetonitrile in water. Prior to the liquid chromatography (LC)-MS measurement, the peptide fractions were resuspended in 0.1% TFA and 2% acetonitrile in water. Chromatographic separation was performed on an Easy-Spray Acclaim PepMap column (50 cm length × 75 μm inner diameter; Thermo Fisher Scientific) at 55 °C by applying 90 min acetonitrile gradients in 0.1% aqueous formic acid at a flow rate of 300 nl/min. An UltiMate 3000 nano-LC system was coupled to a Q Exactive HF-X mass spectrometer via an easy-spray source (all Thermo Fisher Scientific). The Q Exactive HF-X was operated in TMT mode with survey scans acquired at a resolution of 60,000 at m/z 200. Up to 18 of the most abundant isotope patterns with charges 2–5 from the survey scan were selected with an isolation window of 0.7 m/z and fragmented by higher-energy collision dissociation with normalized collision energies of 32, while the dynamic exclusion was set to 35 s. The maximum ion injection times for the survey scan and dual MS (MS/MS) scans (acquired with a resolution of 45,000 at m/z 200) were 50 and 130 ms, respectively. The ion target value for MS was set to 3e6 and for MS/MS was set to 1e5, and the minimum AGC target was set to 1e3.The data were processed with MaxQuant v. 1.6.17.0[91] and the peptides were identified from the MS/MS spectra searched against Uniprot mouse reference proteome (UP000000589) using the built-in Andromeda search engine. Reporter ion MS2-based quantification was applied with reporter mass tolerance = 0.003 Da and min. reporter PIF = 0.75. Cysteine carbamidomethylation was set as a fixed modification and methionine oxidation, glutamine/asparagine deamination, as well as protein N-terminal acetylation, were set as variable modifications. For in silico digests of the reference proteome, cleavages of arginine or lysine followed by any amino acid were allowed (trypsin/P), and up to two missed cleavages were allowed. The false discovery rate (FDR) was set to 0.01 for peptides, proteins, and sites. A match between runs was enabled. Other parameters were used as pre-set in the software. Unique and razor peptides were used for quantification enabling protein grouping (razor peptides are the peptides uniquely assigned to protein groups and not to individual proteins). Reporter intensity corrected values for protein groups were loaded into Perseus v. 1.6.10.0[92]. Standard filtering steps were applied to clean up each dataset: reverse (matched to decoy database), only identified by site, and potential contaminant (from a list of commonly occurring contaminants included in MaxQuant) protein groups were removed. Reporter intensity corrected values were Log2 transformed. Protein groups with all values were kept. Reporter intensity values were then normalized by median subtraction within TMT channels. Student's t-tests were performed on the these values for groups of samples constituting a given dataset. Student's t-test (permutation-based FDR = 0.01, S0 = 0.1) was performed to return protein groups, which levels were statistically significantly changed between the sample groups investigated. This dataset has been deposited to the ProteomeXchange Consortium via the PRIDE repository with the dataset identifier PXD050438.

## Bioinformatic analysis

MMRF CoMMpass study whole-genome DNA sequencing reads were mapped to the GRCh38 human reference genome, using STAR[93] processed and filtered using SAMtools[94]. To identify translocations, significant structural variants were called using SVDetect[95] with default settings. Significant translocations were filtered for those known to be commonly initiated in the Igh locus and genotype. RNA sequencing reads were mapped using STAR split read aligner. RNA sequencing samples of patients genotyped based on exome sequencing as being a DIS3 somatic mutation or a recurrent DIS3 mutation were compared to RNA samples from patients not possessing such a genotype as well as not possessing a germline DIS3 mutation or a mutant DIS3 variant as a minor clone. Reads were counted onto a gene annotation (gencode v34 basic) using FeatureCount (subread package release 2.0.0)[96]. Differential expression RNA-seq analyses were performed as previously using DESeq2 R package[97]. Whole-exome DNA sequencing reads were mapped to the GRCh38 human reference genome, using STAR split read aligner[93], and single nucleotide variants were called using SAMtools and BCFtools pipeline[94]. Healthy tissue control whole-exome DNA sequencing data were mapped and variants that were identified in at least one control sample were defined as germline mutations. The Single Base Substitution (SBS) profiles were calculated using SigProfilerMatrixGenerator[98] and fitted to the COSMIC Mutational Signatures (v3.2) using mmsig[71].

To identify translocations in plasmacytoma DNA samples, mate-pair DNA sequencing data were preprocessed with NXtrim[99]. Both mate-pair and standard TruSeq DNA libraries were mapped by STAR split read aligner. Statistically significant structural variants were called using SVDetect with default settings. All significant structural variants from all sequencing experiments have been concatenated together according to the genotype and merged using bedtools merge to exclude loci involved in more than one structural variant to be analyzed multiple times. These annotations have been used for the meta-analysis. For the filtering to identify those involving an extended region, including the immunoglobulin heavy chain locus, custom scripts were used.

Whole genome sequencing data were used to identify structural variants (SVs) with single-nucleotide breakpoint refinement by Manta (v1.6.0)[100]. Manta breakpoints with precise confidence intervals were extracted along with stretches of microhomology flanking annotations. Filtered breakpoints were formatted into tab-delimited files for microhomology analysis. Microhomology length distributions were visualized in R using histograms and density plots to infer DNA repair signatures related to AID-driven breaks. Analyses employed BEDtools, bcftools, samtools, and UCSC utilities in a Linux environment with a Python 2.7 environment for variant processing.

DamID-seq data were stripped of sequencing adapters using cutadapt and mapped to the mouse reference genome (GRCm38) using STAR aligner. Uniquely mapping reads were counted onto DpnI restriction fragments in the mm10 genome. Count tables were library size corrected. Counts for the 3 replicates of the DamOnly negative control have been averaged and subtracted from all replicates to eliminate the background signal. The annotation of putative PROMPT 3 kb regions upstream and antisense to the known transcription start sites in the gencode release M25. Mean values have been calculated of DIS3 occupancy.

ChIP-seq reads were aligned to the mouse reference genome (GRCm38.p6) using bowtie2. All replicates were pooled and used to call peaks with MACS2 (v2.2.7.1) using IGG control as the background. The obtained tracks and peak annotations were utilized in downstream analysis. Super-enhancer annotations were obtained from publicly

 

available data (GSE6206326)[29]. The coordinates, originally in the mm9 genome assembly, were converted to the GRCm38.p6 genome assembly using the UCSC Genome Browser's liftOver tool.

Meta-analyses of genomic regions were performed using custom scripts employing BWtool. Structural variants hotspots were tested for nucleotide composition similarities and their relation to AID chromatin occupancy or transcription using custom scripts. The AID DNA recognition motifs tracks were prepared by calculating the density of RGYW, GAGCT, GGGGW, and GGGCT sequence occurrence in the GRCm38.p6 mouse reference genome. Two G-quadruplex prediction annotations were performed by scanning the GRCm38.p6 mouse genome for at least four stretches of at least three G separated by up to seven of any nucleotides for a more stringent annotation or separated by 30 of any nucleotides for the less stringent annotation. From these annotations, strand-specific density tracks were prepared and used for meta-analysis. Single-nucleotide variants in plasmacytoma DNA were called using the SAMtools/BCFtools pipeline. Statistically supported variants that were uniquely present in a single tumor sample were defined as somatic. All possible $C \rightarrow T$ transitions and $C \rightarrow G$, $C \rightarrow A$ transversions were identified, and their directionality was assigned to the corresponding genomic strand ( + or − ). Strand-specific RNA-seq signals were then aggregated in a strand specific maner, to assess the strand asymmetry of somatic hypermutation (SHM). Results were then visualized and statistic was performed in R.

To assess the spatial relationship between chromosomal translocation breakpoints and genomic features associated with chromatin activity, we compared their distances to significant ChIP-seq peaks (H3K4me1, H3K4me3, CTCF, Rad21, p300) and super-enhancer annotation. The analysis was conducted using BEDTools. For each breakpoint, the nearest chromatin feature was identified using bedtools closest. Randomized control datasets were generated by shuffling breakpoint coordinates across the genome while preserving chromosomal distribution, using bedtools shuffle. The absolute distances between breakpoints and features in the observed and randomized datasets were compared using the Wilcoxon rank-sum test to assess enrichment or depletion of proximity.

Additionally, the relative distance distributions (fractional distances within feature intervals) were calculated using bedtools reldist. Observed relative distance frequencies were compared to those expected under the random model using Pearson's $\chi^2$ test.

The likelihood of G766R mutation pathogenicity, was assessed using the PON-P2 algorithm[101], with default settings. This machine learning-based classifier categorizes variants into pathogenic, neutral, and unknown classes by utilizing various information, such as evolutionary conservation of sequences, physical and biochemical properties of amino acids, GO annotations, and, if accessible, functional annotations of variation sites.

Raw Hi-C reads were trimmed using TrimGalore (v0.6.7)[102,103]. The fastq files were processed using the Juicer Pipeline (v2.13.07)[104], using default options and aligned to the GRCm38/mm10 genome assembly. The processing was performed without any input for restriction enzyme sites as MicroC uses the Micrococcal Nuclease (MNase) which does not cleave at specific cut-sites.

Raw ligation frequencies were obtained from the.hic files using juicer tool[3] with the following command: juicer dump observed NONE <input.hic > <chr12 > <chr12 > BP 2000 output.txt for each replicate separately. Next, the data was imported into R. The ligation frequency matrix for chromosome 12 was normalized using the Iterative Proportional Fit algorithm (matrix balancing) as described previously[105] (https://pekowskalab.nencki.edu.pl/research/the-code).

Publicly available data from the following publications were used: ChIP-seq AID SRA: SRP003605[23]; ChIP-seq H3K4me3, H3K4me1, H3K27ac, p300, CTCF and Rad21 SRA: SRP075985[64]; EXOSC3 KO RNA-seq SRA: SRP042355[7]; GRO-seq B-cells and CH12F3-2A SRA: SRP193758[30]; Super enhacer annotation: GSE62063[29].

## Reporting summary

Further information on research design is available in the Nature Portfolio Reporting Summary linked to this article.

## Data availability

The sequencing data have been deposited in NCBI's Gene Expression Omnibus (GEO) under the accession code GSE155631. The raw sequencing data have been deposited in the Sequence Read Archive (SRA) under the accession code SRP275679, associated with BioProject PRJNA650522. The mass spectrometry proteomics data have been deposited in the ProteomeXchange Consortium via the PRIDE repository under the accession code PXD050438. Source data are provided with this paper.

## Code availability

All custom scripts used for analyses have been deposited in Mendeley Data and are publicly available at: 10.17632/7tdgwkzmnr.1.

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

## Acknowledgements

We thank Justyna Chlebowska, Jakub Gruchota, Kamila Kłosowska-Kosicka, Dorota Adamska, Michał Kamiński, Radosław Salamon, Katarzyna Prokop, and Monika Kusio-Kobiałka for help with selected experiments, Maria Anna Ciemerych-Litwinienko, Dominika Nowis, Jakub Gołąb, Joanna Kufel, Katarzyna Matylla-Kulińska for attentive readings of the manuscript and all Andrzej Dziembowski lab members for fruitful discussions. We thank the expert support of the Mouse Clinic Institute (Illkirch) in mouse construction and handling. Efforts of the MM Research Foundation (MMRF) and centers that contributed to the CoMMpass study are also acknowledged. All mice lines were genotyped by the Genome Engineering Facility (GEF), part of IIMCB IN-MOL-CELL Infrastructure (RRID: SCR_021630) funded by the European Union – NextGenerationEU under National Recovery and Resilience Plan, Horizon Europe (Project 101059801 - RACE) and RACE-PRIME project carried out within the IRAP program of the Foundation for Polish Science co-financed by the European Union under the European Funds for Smart Economy 2021-2027 (FENG) IIMCB. This work was mainly supported by grant funding from the National Science Center (NCN) to AD (UMO-2016/22/A/NZ4/00380; UMO-2013/10/M/NZ4/00299) and TK (UMO-2019/32/C/NZ2/00558). This research was co-supported by funding from the European Union's Horizon 2020 research and innovation program (grant agreement no. 810425). Work in BS laboratory was supported by INCA PRTK 2021-025, USIAS, ANR-10-IDEX-0002, ANR 20-SFRI-0012, ANR-17-EURE-0023. Work in the AP laboratory is funded by the Dioscuri Grant [Dioscuri is a program initiated by the Max Planck Society (MPG), jointly managed with the National Science Center in Poland (NCN), and mutually funded by the Polish Ministry of Education and Science and the German Federal Ministry of Education and Research (BMBF)]; by the OPUS17 (UMO-2019/33/B/NZ2/02437), OPUS22 (UMO-2021/43/B/NZ2/02934), and Sonata Bis 11 (UMO-2021/42/E/NZ2/00392) grants from the NCN; and by the EMBO Installation Grant.

## Author contributions

T.K.: conceptualization, all bioinformatic analysis, and experimental work; O.G., M.M., N.D., K.K., A.S.C., M.J.S., E.P.O., and M.N.: experimental work; A.P. and D.C.: bioinformatic analysis B.S.: conceptualization and knock-in mice; AD: conceptualization, supervision, original draft preparation.

## Competing interests

The authors declare no competing interests.

## Additional information

**Ethical issues** All procedures were approved by the First Local Ethical Committee in Warsaw affiliated at the University of Warsaw, Faculty of Biology (approval numbers WAW/092/2016, WAW/177/2016, WAW/642/2018). Housing in animal facilities was performed in conformity with local and European Commission regulations under the control of veterinarians and with the assistance of trained technical personnel.

Tomasz M. Kuliński[1,2] ✉, Olga Gewartowska [3,4], Mélanie Mahé[5,6], Karolina Kasztelan[1,2], Nina Durys[2], Anna Stroynowska-Czerwińska [7], Marta Jedynak-Slyvka[8,9,10], Ewelina P. Owczarek[1], Debadeep Chaudhury [7], Marcin Nowotny[8], Aleksandra Pękowska [7], Bertrand Séraphin [5,6] & Andrzej Dziembowski [1,2,3] ✉

[1]Laboratory of RNA Biology, International Institute of Molecular and Cell Biology, Warsaw, Poland. [2]Institute of Biochemistry and Biophysics, Polish Academy of Sciences, Warsaw, Poland. [3]Faculty of Biology, University of Warsaw, Warsaw, Poland. [4]Genome Engineering Facility, International Institute of Molecular and Cell Biology, Warsaw, Poland. [5]Institut de Génétique et de Biologie Moléculaire et Cellulaire, UMR, 7104, and Centre National de Recherche Scientifique, Illkirch, France. [6]Institut National de Santé et de Recherche Médicale, U964, and Université de Strasbourg, Illkirch, France. [7]Dioscuri Center for Chromatin Biology and Epigenomics, Nencki Institute of Experimental Biology, Polish Academy of Sciences, Warsaw, Poland. [8]Laboratory of Protein Structure, International Institute of Molecular and Cell Biology, Warsaw, Poland. [9]International Institute of Molecular Mechanisms and Machines, Polish Academy of Sciences, Flisa 6, Warsaw, Poland. [10]Present address: Technical University of Denmark, Centre for Diagnostics, Department of Health Technology, Henrik Dams Allé, 2800 Kgs, Lyngby, Denmark. ✉e-mail: tkulinski@iimcb.gov.pl; adziembowski@iimcb.gov.pl

