## [Transparent Peer Review file · Nature Communications]

DIS3 Mutations Enhance AID-Driven Translocations During B-Cell Activation, Promoting Transformation to Multiple Myeloma

Corresponding Author: Professor Andrzej Dziembowski

Version 0:

Reviewer comments:

Reviewer #1

(Remarks to the Author)

Activation Induced cytidine Deaminase (AID) catalyzes Class switch recombination (CSR) and somatic hypermutation (SHM) to promote antibody gene diversification. How AID identified the IgH locus to promote antibody gene specific SHM without mutating various other oncogenes in the B cell genome is an important and exciting topic of research. Previous studies have established that AID functions by associating with the exoribonuclease RNA exosome complex with both associating with a stalled RNA polymerase II Complex. While stalled RNA polIII recruits AID, the RNA decay activity of RNA exosome decays the RNA associated with the stalled polIII to resolve transcription stalling and stimulate RNA polIII elongation and provide single strand DNA substrates for AID. Loss of RNA exosome activity lead to reduced AID activity on the IGH locus while catalytically dead RNA pol II (DIS3-depleted) lead to loss of CSR and increase in IgH chromosomal translocation(Laffleur et al , 2021). In these studies, multiple transcriptional effects on the noncoding genome of B cells were observed due to RNA exosome loss of function. In this exciting study, the authors have identified a gain of function DIS3 mutant (DIS3766R) that promotes IgH chromosomal translocation that are associated with B cell transformation due to increased association with AID on chromatinized DNA, without affecting other aspects of RNA exosome function on B cell transcriptome. This study provides important evidence of the role of chromatin associated RNA in recruiting AID to its target sites and expands our knowledge of the mechanism of AID targeting. The identification of the DIS3766R mutation, that separates RNA exosome function in general cellular RNA degradation from its role in AID oncogenic activity, will be a widely used information in the AID targeting and B cell malignancy fields. Although many oncogenic mutations are identified in various B cell cancers, rarely are they followed up properly to establish their mechanism in malignant transformation. In this regard, this study is unique. The study has been done with significant experimental I propose publication of this manuscript. Some revisions, as suggested below, may improve the manuscript and clarify some important points.

1. The authors could reconsider the title of the manuscript. The association of the mutation with multiple myeloma is important but could be misleading. These AID associated mutations, most likely, occur in the germinal center and persist in the plasma B cells (that are associated with MM). CSR is also unperturbed for the specific mutation being investigated here. Thus, focusing on the main point- Oncogenic DIS3 mutation supports AID activity to cause chromosomal translocations- maybe more accurate. This is a suggestion and authors can decide what they think is the best.
2. In fig. 1F, the authors show increased R-loops over promoters. Analyses of R-loop status at promoters of AID off-target genes would be better to show here (as an additional ub figure), with an example of one IGV track of the data that reflects the quality of the DRIP-seq.
3. In figure 3, the authors demonstrate the translocation in the DIS3+/G766R mutation. What are the targets of these translocations and are there microhomologies in the translocation junctions. Are there any evidence of asymmetric SHM at the translocation breakpoints.
4. In figure 6, authors suggest that stalled DIS3G766R mutant protein occur on chromatin associated RNAs. Are these the same RNAs that accumulate at R-loops at the promoters? How are the authors correlating stalling DIS3 on chromatinized RNA with them being R-loop driving.
5. Finally, in the discussion that authors could also refer to the mechanism of stabilization of single strand DNA through secondary DNA structures (PMID: 31509023) and AID phosphorylation and association with single strand DNA binding RPA (PMID: 19442251). These are the fundamental mechanism that DIS3 may collaborate with to promote oncogenic translocations.

Reviewer #2

(Remarks to the Author)

In this manuscript, Kulinsky et al examine the role of DIS3, a key component RNA exosome complex in driving mutations and translocations in multiple myeloma. The authors knocked-in a G766R mutation, a clinically relevant DIS3 variant, into the endogenous DIS3 locus. They find that while mice homozygous for the mutation is embryonically lethal, heterozygous DIS3

(G766R) mice develop normally, allowing them to examine the contribution of this mutation to cellular transformation. The authors find that the heterozygous DIS3(G766R) mutations drive murine plasmacytomas in a pristane-induced model, and whole-genome sequencing revealed increased IgH-dependent chromosomal translocations. They also find that mutated DIS3 accumulates on chromatin bound RNA, and especially at sites that are known off-target AID sites. Employing a CH12 B lymphoma line, the authors demonstrate that there is increased DNA damage in cells expressing DIS3(G766R), and they also demonstrate increased physical proximity of Igh and protooncogenes. These results prompted the authors to propose a model wherein DIS3 mutations enhance AID promiscuity to drive IgH translocations and multiple myeloma development.

The mechanisms that protect the B cell genome from AID-induced collateral damage is of much relevance to our understanding of B cell transformation. The role of the RNA exosome complex in AID targeting has been extensively studied in the context of class switching and somatic hypermutation, but how (and if) mutations in RNA exosome proteins found in patients with multiple myeloma contribute to cellular transformation is poorly understood. Thus, generating a mouse model that recapitulates some aspects of the human disease is of potential interest. However, the authors need to address several issues to strengthen the work.

1. The major claim of the authors based on the data presented is that the translocations are AID-dependent. However, other than indirect evidence based on sequence motifs etc, there is no data showing that this is indeed the case. The authors should absolutely provide data to show that the translocations they pick up are indeed AID-dependent. AID-deficient mice are generally available and there are multiple ways to delete AID from cells/cell lines. Without this direct experimental evidence, it is difficult to evaluate the major findings of the study.

2. The authors should do a thorough analysis of the germinal center response of the knock-in mice. Are there altered GC B cell responses at homeostasis and after immunization, altered LZ:DZ ratio, effects on affinity maturation?

3. The manipulated CH12 cells (Fig 6) are very poorly characterized. Are these clones of CH12 cells that have the KI mutation? Or are pools being analyzed? What are the DIS3 protein levels? What happens when these cells are stimulated? Do they switch like the more widely used CH12F3-2B subclone? For Fig 3D, how do the authors know that they are indeed detecting AID? Without these control data, it is nearly impossible to evaluate the data presented.

Reviewer #3

(Remarks to the Author)

→ In this manuscript the authors examine the functional effects of a heterozygous hypomorphic Dis3 mutation in a novel mouse model. They find that the mice, particularly males, are smaller, the B-cell less proliferative and susceptible to pristane induced plasmacytomas with increased chromosomal translocations. This is consistent with their provocative ref57 which postulates that Dis3 mutations are an early event which predispose to IgH translocations but are subsequently selected against because of the anti-proliferative effects. The current manuscript fully supports this novel and original hypothesis, but leaves a number of unanswered questions. There are a large number of experiments with excellent bioinformatic analyses.

There are some minor points:

Table S12 and S13 are supposed to list DIS3G766R/+ and WT plasmacytoma translocations respectively. I could not identify the IgH translocations shown in the Circos plots in Figure 3D, and I wonder if the wrong tables were uploaded. It would be helpful to highlight in the table the IgH translocations shown in the Circos plots. It is not clear how many plasmacytomas were sequenced, and whether each had a unique IgH translocation, only some had an IgH translocations, as in humans, or if some had multiple IgH translocations. Were cell lines established? Are the tumor and translocations polyclonal, clonal or subclonal? Are any translocations recurrent? Do they drive the expression of adjacent oncogenes? DO they appear to be driver or passenger mutations? It is somewhat surprising that Igh-Myc translocations were not observed as they are almost universal in other pristane-induced plasmacytoma models.

Although not critical to support their main hypothesis additional characterization of DIS3G766R/+ would be nice. C57Bl/6 spontaneously develop monoclonal gammopathy with age. Is this accelerated in DIS3G766R/+? Is there any change in isotype distribution in DIS3G766R/+ compared to WT?

The English can be improved:

“Taken together, by introducing Dis3G766R mutation in a heterozygous setting, we recapitulate the clinical markup of MM cells”

I am not sure what they are trying to say, but the mouse does not reproduce clinical features of human MM (Anemia, bone disease, renal lesions).

Version 1:

Reviewer comments:

Reviewer #1

(Remarks to the Author)

The authors have addressed many of my queries with experimental data and well thought out arguments.

This is an important study that clearly shows the role of DIS3 in stimulating AID's DNA deamination activity on chromatinized DNA. It will be well followed in the B cell malignancy, RNA biology and antibody somatic hypermutation fields. I propose publication.

Reviewer #2

(Remarks to the Author)

The authors have addressed all the issues raised by this reviewer. This is now an important contribution to the field.

Reviewer #3

(Remarks to the Author)

The authors have responded constructively to the previous critique, substantially improving the manuscript. The findings reported are highly novel and significant.

The following minor points remain.

Figure S1E. Serum Protein Electrophoresis in wildtype C57Bl/6 is not expected to show a change in the protein fractions reported in this figure. The authors should consult with someone proficient at analyzing SPEPs (e.g., a hematologist or lab technician). The abnormality seen with age in C57Bl/6 mice is a very faint monoclonal band in the beta or, more commonly, the gamma globulins. It is not of sufficient size to increase the level of total globulins in the fraction. About 30% of 1 year old C57Bl/6 mice will have a faint monoclonal band detected on SPEP.

Table S12,S13,S14,S15. These bedpe files have a terminal column with numerical entries such as ;3/4;5/6. I suspect this is supposed to be left split reads/discordant reads; right split reads/discordant reads supporting the presence of the SV. It is not clear to me why all 4 numbers reported for each SV are identical, please check that this is correct. Furthermore, they appear to use a very low cutoff to report SV (3 reads supporting SV). This will be more sensitive to subclonal SV, but also introduces a lot of noise. I suspect a higher cutoff will provide a cleaner analysis, although they have convincingly made their point with their current analysis (if less than ideal).

One of the IgH SV they report in the G766R Dis3 plasmacytomas has a well-supported breakpoint (67 reads) 170kb from Tnfrsf17 (Bcma), a gene that is sometimes similarly dysregulated in human myeloma. It would be nice if they can demonstrate over-expression of Tnfrsf17 in the sample with this translocation.

I am unable to find the MYC SV shown in Figure S6E in Table S12. The SV in S6B,C,D,E do not overlap DRIPseq peaks as in Figure 4K and S6A. It is not clear how they support their argument.

I am unable to access the GSE155631. Be sure it contains the older mate-pair, and newer whole genome data. In my experience mouse whole genome and mate pair DNA sequencing data is submitted to SRA, not GEO.

Leif Bergsagel

Kuliński et al. - point-by-point response to the editors and reviewers' comments

We thank the reviewers for their overall positive opinion of our manuscript. We have addressed all the issues requested, which substantially improved the manuscript.

The major limitation of the initial version of our manuscript raised by the reviewers was the lack of direct evidence of the AID involvement in the DIS3-dependent translocations. In a new version, we have introduced new crucial experiments and bioinformatic analysis, which significantly contributed to our statement that the mechanism of oncogenic action of DIS3 mutation is indeed AID-dependent.

First, we demonstrate that in AID-expressing CH12F3.2a cells, a population of cells exists that, upon activation, exhibits an increased number of double-stranded DNA breaks. This number is slightly increased upon expression of the MM DIS3 mutant allele. Notably, this genome instability readout is strictly AID-dependent, as, regardless of the expression of the DIS3 variant, a very small population with increased dsDNA breaks is observed in AID-KO CH12F3.2a cells.

Second: Typically, during somatic hypermutation (SHM), mutations are predominantly found on the non-transcribed strand of the transcription bubble, a consequence of asymmetric DNA lesion deposition by AID. In contrast, class switch recombination (CSR), which depends on the generation of double-stranded DNA breaks, requires AID activity on both DNA strands. It has been postulated that DIS3 contributes to the targeting of AID, particularly to the template strand. Following the suggestion of Reviewer 1, we investigated the effect of DIS3 mutation on the accumulation of somatic mutations in plasmacytoma samples, utilizing both DNA sequencing and strand-specific RNA sequencing data from B cells activated *in vitro*. Strikingly, we observe the expected strand asymmetry of mutations in wild-type mice; however, in the case of the DIS3^{G766R} mutant, this asymmetry is lost, and mutations are equally distributed on both strands of the transcription bubble. This is precisely the pattern one would expect in the context of increased AID-dependent translocations, which depend on double-stranded DNA breaks.

REVIEWERS COMMENTS

Reviewer #1 (Remarks to the Author):

Activation Induced cytidine Deaminase (AID) catalyzes Class switch recombination (CSR) and somatic hypermutation (SHM) to promote antibody gene diversification. How AID identified the IgH locus to promote antibody gene specific SHM without mutating various other oncogenes in the B cell genome is an important and exciting topic of research. Previous studies have established that AID functions by associating with the exoribonuclease RNA exosome complex with both associating with a stalled RNA polymerase II Complex. While stalled RNA polIII recruits AID, the RNA decay activity of

RNA exosome decays the RNA associated with the stalled polII to resolve transcription stalling and stimulate RNA polII elongation and provide single strand DNA substrates for AID. Loss of RNA exosome activity lead to reduced AID activity on the IGH locus while catalytically dead RNA pol II (DIS3-depleted) lead to loss of CSR and increase in IgH chromosomal translocation (Laffleur et al , 2021). In these studies, multiple transcriptional effects on the noncoding genome of B cells were observed due to RNA exosome loss of function. In this exciting study, the authors have identified a gain of function DIS3 mutant (DIS3766R) that promotes IgH chromosomal translocation that are associated with B cell transformation due to increased association with AID on chromatinized DNA, without affecting other aspects of RNA exosome function on B cell transcriptome. This study provides important evidence of the role of chromatin associated RNA in recruiting AID to its target sites and expands our knowledge of the mechanism of AID targeting. The identification of the DIS3766R mutation, that separates RNA exosome function in general cellular RNA degradation from its role in AID oncogenic activity, will be a widely used information in the AID targeting and B cell malignancy fields. Although many oncogenic mutations are identified in various B cell cancers, rarely are they followed up properly to establish their mechanism in malignant transformation. In this regard, this study is unique. The study has been done with significant experimental I propose publication of this manuscript. Some revisions, as suggested below, may improve the manuscript and clarify some important points.

We thank the reviewer for the positive opinion about our paper

1. The authors could reconsider the title of the manuscript. The association of the mutation with multiple myeloma is important but could be misleading. These AID associated mutations, most likely, occur in the germinal center and persist in the plasma B cells (that are associated with MM). CSR is also unperturbed for the specific mutation being investigated here. Thus, focusing on the main point- **Oncogenic DIS3 mutation supports AID activity to cause chromosomal translocations**- maybe more accurate. This is a suggestion and authors can decide what they think is the best.

We thank the reviewer for the thoughtful suggestion regarding the manuscript title. In response, we have revised the title to:

"DIS3 Mutations Enhance AID-Driven Translocations During B-Cell Activation, Promoting Transformation to Multiple Myeloma."

This revised title highlights the central mechanistic finding — that DIS3 mutations potentiate AID-dependent genomic instability during B-cell activation — while avoiding the implication that class switch recombination (CSR) is functionally impaired. We have chosen to retain the reference to "multiple myeloma" to emphasize the disease context and clinical relevance of our study.

2. In fig. 1F, the authors show increased R-loops over promoters. Analyses of R-loop status at promoters of AID off-target genes would be better to show here (as an additional ub figure), with an example of one IGV track of the data that reflects the quality of the DRIP-seq.

We agree with the reviewer that the suggested data would serve as valuable supporting controls. However, we believe that incorporating these additional panels into Figure 1 would disrupt the logical flow of the main text and figure narrative. The relevant analyses, including R-loop profiles over AID off-target genes and representative IGV tracks demonstrating DRIP-seq quality, are already presented in Supplementary Fig. S6, which we believe appropriately complements the main figure without redundancy.

3. In figure 3, the authors demonstrate the translocation in the DIS3^{+/G766R} mutation. What are the targets of these translocations and are there microhomologies in the translocation junctions.

Microhomology at translocation junctions:

To analyze potential microhomologies at translocation junctions, we initially faced limitations due to the DNA sequencing data, particularly mate-pair libraries, which often prevent precise breakpoint identification. To address this, we curated the data to identify exact chromosomal breakpoints, representing a subset of mapped translocations. Homologous sequences immediately adjacent to these breakpoints were then annotated. Nucleotide-resolution mapping revealed microhomology-mediated repair in both genotypes, consistent with AID-dependent B-cell neoplasms and MM79,80. Notably, DIS3^{G766R/+} cells exhibited slightly longer microhomologies (7 bp vs. 5 bp in WT), suggesting increased reliance on alternative end joining (a-EJ).

Translocation targets:

Mapped translocations involve immunoglobulin loci and known AID-targeted genomic regions, with many partners overlapping highly transcribed genes and super-enhancer regions in B cells, consistent with previous observations in plasmacytomas and multiple myeloma. For example, Myc translocations were identified and are shown in the S6. However, given the sequencing depth and mate-pair library protocol, our approach was not optimized for exhaustive detection of clonal driver rearrangements. Instead, the pipeline captures a broad spectrum of statistically supported events, reflecting the overall translocation landscape and sequence specificity associated with the DIS3^{G766R} mutation. Accordingly, our study emphasizes the mutator phenotype and the mechanistic consequences of DIS3 dysfunction rather than the physiological impact of specific deregulated signaling pathways.

Is there any evidence of asymmetric SHM at the translocation breakpoints?

We thank the reviewer for this insightful suggestion, which, as already indicated, allowed us to strengthen the support for AID dependence of translocations observed in $DIS3^{G766R/+}$ mice.

Translocation breakpoints themselves do not provide strand-specific information relative to transcription. To address this limitation, we leveraged both plasmacytoma DNA sequencing and strand-specific RNA sequencing from in vitro-activated B cells. Rather than focusing solely on breakpoints, we performed a genome-wide analysis of somatic mutations.

Somatic SNVs were identified in plasmacytoma DNA, with all C→T transitions and C→G C→A transversions cataloged and assigned to the corresponding genomic strand (+ or -). Strand-specific RNA-seq signals were then aggregated in a strand-aware manner to evaluate SHM asymmetry.

This approach revealed clear differences in strand-specific mutation patterns between $DIS3$ genotypes, as described in our response to the editor and now shown in Fig. 6I of the revised manuscript. In the context of $DIS3^{G766R/+}$, AID loses most of its preference for the coding strand, exhibiting nearly equal activity on both strands which fits into our proposed model.

4. In figure 6, authors suggest that stalled $DIS3^{G766R}$ mutant protein occur on chromatin associated RNAs. Are these the same RNAs that accumulate at R-loops at the promoters? How are the authors correlating stalling $DIS3$ on chromatinized RNA with them being R-loop driving.

In the original version of the manuscript, the distinction between co-transcriptional RNA and RNA associated with chromatin was not sufficiently clear and may have caused confusion. At the same time, due to the already quite complex nature of the manuscript, we did not want to introduce too many distinctions. In this revised version, we more clearly state that R-loop-engaged RNAs represent only a fraction of DIS3 substrates at the chromatin (Figure 1), including those associated with chromatin. It starts with the initial RNA-seq and DRIP seq data, and is also indicated in the DamID-related data presented Figure 6.

5. Finally, in the discussion that authors could also refer to the mechanism of stabilization of single strand DNA through secondary DNA structures (PMID: 31509023) and AID phosphorylation and association with single strand DNA binding RPA (PMID: 19442251). These are the fundamental mechanism that DIS3 may collaborate with to promote oncogenic translocations.

We thank the reviewer for this excellent suggestion. We have incorporated discussion of these mechanisms, referencing the stabilization of single-stranded DNA through secondary DNA structures and the role of AID phosphorylation and its interaction with RPA. Including these aspects has made our discussion of the proposed model more comprehensive, providing additional mechanistic context for how mutant DIS3 may cooperate with established AID regulatory pathways to promote oncogenic translocations.

Reviewer #2 (Remarks to the Author):

In this manuscript, Kulinsky et al examine the role of DIS3, a key component RNA exosome complex in driving mutations and translocations in multiple myeloma. The authors knocked-in a G766R mutation, a clinically relevant DIS3 variant, into the endogenous DIS3 locus. They find that while mice homozygous for the mutation is embryonically lethal, heterozygous DIS3(G766R) mice develop normally, allowing them to examine the contribution of this mutation to cellular transformation. The authors find that the heterozygous DIS3(G766R) mutations drive murine plasmacytomas in a pristane-induced model, and whole-genome sequencing revealed increased IgH-dependent chromosomal translocations. They also find that mutated DIS3 accumulates on chromatin bound RNA, and especially at sites that are known off-target AID sites. Employing a CH12 B lymphoma line, the authors demonstrate that there is increased DNA damage in cells expressing DIS3(G766R), and they also demonstrate increased physical proximity of Igh and protooncogenes. These results prompted the authors to propose a model wherein DIS3 mutations enhance AID promiscuity to drive IgH translocations and multiple myeloma development.

The mechanisms that protect the B cell genome from AID-induced collateral damage is of much relevance to our understanding of B cell transformation. The role of the RNA exosome complex in AID targeting has been extensively studied in the context of class

switching and somatic hypermutation, but how (and if) mutations in RNA exosome proteins found in patients with multiple myeloma contribute to cellular transformation is poorly understood. Thus, generating a mouse model that recapitulates some aspects of the human disease is of potential interest.

We thank the reviewer for the interest in our work.

However, the authors need to address several issues to strengthen the work.

1. The major claim of the authors based on the data presented is that the translocations are AID-dependent. However, other than indirect evidence based on sequence motifs etc, there is no data showing that this is indeed the case. The authors should absolutely provide data to show that the translocations they pick up are indeed AID-dependent. AID-deficient mice are generally available and there are multiple ways to delete AID from cells/cell lines. Without this direct experimental evidence, it is difficult to evaluate the major findings of the study.

We thank the reviewer for this essential comment, which helped us strengthen the key mechanistic conclusion of the study. As requested, in the revised version of the manuscript we now provide direct experimental evidence demonstrating that the genome instability caused by DIS3 mutations is AID-dependent.

To address this point, we employed the CH12F3-2A cell line in both AID-proficient (AID^{+/+}) and AID-deficient (AID^{-/-}) backgrounds. Cells were transduced with lentiviral constructs expressing either DIS3^{WT} or DIS3^{G766R}, and γ -H2AX levels were quantified as a marker of DNA double-strand breaks. Consistent with our primary B-cell data, DIS3^{G766R} expression caused a statistically significant increase in γ -H2AX signal compared to DIS3^{WT} in AID^{+/+} cells (Fig. 6H). Crucially, this effect was strictly AID-dependent, as AID^{-/-} cells expressing DIS3^{G766R} did not exhibit elevated γ -H2AX levels. These findings demonstrate that AID activity is required for the accumulation of DNA damage in the presence of mutant DIS3. No differences were observed in non-activated CH12F3-2A cells regardless of genotype.

Full experimental details, including biological replicates and representative controls are presented in Figure S8B.

2. The authors should do a thorough analysis of the germinal center response of the knock-in mice. Are there altered GC B cell responses at homeostasis and after immunization, altered LZ:DZ ratio, effects on affinity maturation?

We thank the reviewer for this important point. We have cytometrically analyzed the germinal center (GC) response in both naïve and immunized $DIS3^{+/G766R}$ and wild-type mice. Following immunization, we observed an increase in the GC B-cell population in the spleens in both genotypes. However, we did not detect any significant difference in the LZ:DZ (light zone to dark zone) ratio between genotypes, indicating that the organization of the GC response remains intact. These results are now included in the

3. The manipulated CH12 cells (Fig 6) are very poorly characterized. Are these clones of CH12 cells that have the KI mutation? Or are pools being analyzed? What are the DIS3 protein levels? What happens when these cells are stimulated? Do they switch like the more widely used CH12F3-2B subclone?

We thank the reviewer for this valuable comment. Indeed, the initial figure description of the CH12 cell line was imprecise, and we appreciate the opportunity to clarify this point. The cell line used in our experiments is the CH12F3-2A clone, which is precisely the commonly used B-cell model referred to by the reviewer. This clone is widely employed

to study class switch recombination (CSR) and B-cell activation. The figure label has now been corrected accordingly, and the Methods section has been updated to describe the cell line accurately.

In our experiments, we analyzed pooled populations of CH12F3-2A cells stably expressing either wild-type or G766R DIS3–Dam fusions; these were not clonally selected. The functionality of the CH12F3-2A cells was verified, including their ability to undergo class switching upon cytokine stimulation, comparable to the more widely used CH12F3-2B subclone. These experimental details are now clearly specified in both the figure legend and the Methods section.

Regarding the DIS3 protein expression, for DamID experiments, it is essential that the Dam-fusion proteins are expressed at very low levels. To achieve this, we utilized leaky expression from an uninduced inducible promoter, resulting in protein levels that are below the detection limit of Western blotting and immunofluorescence microscopy. This low expression is critical, as Dam is a highly active enzyme, and even moderate expression can cause non-specific, saturating methylation. In our system, the fusion protein was expressed from the pLgw V5-EcoDam vector under the minimal heat-shock promoter, ensuring minimal activity. EcoDam specifically methylates adenine in GATC sequences, a modification not naturally found in mammalian genomes, allowing precise mapping of protein–DNA interactions. The specificity of the obtained DamID signal was validated by digestion of negative control samples, confirming that the detected methylation reflected targeted binding events.

For Fig 3D, how do the authors know that they are indeed detecting AID? Without these control data, it is nearly impossible to evaluate the data presented.

We assume that this comment refers to Fig. 6D. All required controls mentioned by the reviewer are present in Supplementary Fig. S8A (which corresponds to the previous version of Fig. 6F). The misleading reference in the Fig. 6D legend has now been corrected to reflect what is accurate.

Reviewer #3 (Remarks to the Author):

In this manuscript, the authors examine the functional effects of a heterozygous hypomorphic Dis3 mutation in a novel mouse model. They find that the mice, particularly males, are smaller, the B-cell less proliferative and susceptible to pristane induced plasmacytomas with increased chromosomal translocations. This is consistent with their provocative ref57 which postulates that Dis3 mutations are an early event which predispose to IgH translocations but are subsequently selected against because of the anti-proliferative effects. The current manuscript fully supports this novel and original hypothesis, but leaves a number of unanswered questions. There are a large number of

experiments with excellent bioinformatic analyses.

We thank the reviewer for their thoughtful and positive assessment of our work. We appreciate the recognition of the novelty and significance of our findings, as well as the efforts we made in our experimental and bioinformatic analyses.

There are some minor points:

Table S12 and S13 are supposed to list DIS3G766R/+ and WT plasmacytoma translocations respectively. I could not identify the IgH translocations shown in the Circos plots in Figure 3D, and I wonder if the wrong tables were uploaded.

We thank the reviewer for pointing this out. We have carefully checked and confirmed that the IgH translocations shown in the Circos plots of Figure 3D are present in the original BEDPE files. To improve clarity, we have now uploaded two additional BEDPE files containing only filtered translocations originating from the IgH locus:

- Table S14: BEDPE file with DIS3G766R/+ plasmacytoma translocations originating from the IgH locus.
- Table S15: BEDPE file with DIS3 WT plasmacytoma translocations originating from the IgH locus.

These new files should make it easier to directly identify the IgH translocations visualized in Figure 3D

It would be helpful to highlight in the table the IgH translocations shown in the Circos plots. It is not clear how many plasmacytomas were sequenced, and whether each had a unique IgH translocation, only some had an IgH translocations, as in humans, or if some had multiple IgH translocations. Were cell lines established? Are the tumor and translocations polyclonal, clonal or subclonal?

Plasmacytomas isolated from five individual mice were subjected to sequencing—three from DIS3^{G766R/+} animals and two from DIS3^{+/+} controls. The translocation profiles observed were largely subclonal /polyclonal in nature, consistent with the heterogeneous composition of primary tumor samples.

Are any translocations recurrent? Do they drive the expression of adjacent oncogenes? DO they appear to be driver or passenger mutations? It is somewhat surprising that Igh-Myc translocations were not observed as they are almost universal in other pristane-induced plasmacytoma models.

None of the identified translocations were strictly recurrent; however, several genomic regions, most notably the Igh locus, acted as translocation hotspots, repeatedly linking to distinct and sequence-unrelated loci across samples.

Given the sequencing depth and the mate-pair library protocol used, our approach was not optimized for exhaustive detection of clonal driver rearrangements. Instead, the analytical pipeline was designed to robustly capture a broad spectrum of statistically

supported translocation events, including subclonal ones, to reflect the overall translocation landscape and sequence specificity associated with the $DIS3^{G766R}$ mutation. Our study thus focuses on the mutator phenotype and mechanistic implications of $DIS3$ dysfunction rather than on the oncogenic potential or physiological impact of specific rearrangements.

Although canonical Igh – Myc translocations, frequently observed in BALB/c plasmacytoma models, were not detected in our dataset, we did, however, identify translocations involving the Myc locus (Fig. S6). The absence of classical Igh – Myc events can stem from technical factors related to sequencing depth, mate-pair library structure, or the necessity to filter repetitive or blacklist regions during data processing.

Although not critical to support their main hypothesis additional characterization of $DIS3^{G766R/+}$ would be nice. C57Bl/6 spontaneously develop monoclonal gammopathy with age. Is this accelerated in $DIS3^{G766R/+}$?

We thank the reviewer for this suggestion. As demonstrated in supplementary Fig. S1E, we examined the proteinograms in $DIS3^{+/+}$ and $DIS3^{G766R/+}$ mice. At the ages analyzed, we did not observe any evidence for acceleration of monoclonal gammopathy compared to wild-type C57Bl/6 mice. While C57Bl/6 mice can spontaneously develop monoclonal gammopathy with age, our data indicate that the $DIS3^{G766R/+}$ mutation does not markedly accelerate this process. We note, however, that we did not perform an extensive clinical characterization of gammopathy, as this was beyond the primary focus of the current study.

Figure S3. (...) (H) No significant changes in frequencies of class switching in $Dis3^{+/+}$ and $DIS3^{G766R/+}$ day 0, day 3 and day 5 in vitro activated B-cells, as determined by the analysis of immunoglobulin heavy and light chain classes expression in RNA-seq experiments.

Is there any change in isotype distribution in $DIS3^{G766R/+}$ compared to WT?

We did not detect any genotype-dependent differences in isotype distribution. To address this question, we analyzed the expression of immunoglobulin heavy- and light-chain classes in in vitro-activated primary B cells collected at days 0, 3, and 5 post-activation using RNA-seq. The relative expression levels of each isotype were comparable between $DIS3^{+/+}$ and $DIS3^{G766R/+}$ cells at all examined time points (Fig. S3H). These results, together with, indicate that the $DIS3^{G766R}$ mutation does not alter class-switch recombination efficiency or isotype usage.

The English can be improved:

“Taken together, by introducing $Dis3^{G766R}$ mutation in a heterozygous setting, we recapitulate the clinical markup of MM cells”

I am not sure what they are trying to say, but the mouse does not reproduce clinical features of human MM (Anemia, bone disease, renal lesions).

We agree with reviewer #3 that the statement was misleading and has been changed in the revised version of the manuscript.

Point-by-Point Response to Reviewers

We thank the Reviewers for their positive assessments and constructive feedback, which have greatly strengthened the manuscript. We are pleased that Reviewers #1 and #2 find the work ready for publication. We also sincerely appreciate Prof. Leif Bergsagel (Reviewer #3) for sharing his name and for his detailed, expert input from a clinical perspective, which has been invaluable both in this round and in previous reviews. Below, we address all points raised.

Reviewer #1 (Remarks to the Author):

The authors have addressed many of my queries with experimental data and well thought out arguments. This is an important study that clearly shows the role of DIS3 in stimulating AID's DNA deamination activity on chromatinized DNA. It will be well followed in the B cell malignancy, RNA biology and antibody somatic hypermutation fields. I propose publication.

Response:

We sincerely thank Reviewer #1 for this enthusiastic endorsement and for recognizing the study's impact across these key fields.

Reviewer #2 (Remarks to the Author):

The authors have addressed all the issues raised by this reviewer. This is now an important contribution to the field.

Response:

We are grateful to Reviewer #2 for confirming that all concerns have been resolved and for highlighting the work's contribution.

Reviewer #3 (Remarks to the Author):

The authors have responded constructively to the previous critique, substantially improving the manuscript. The findings reported are highly novel and significant.

The following minor points remain.

Figure S1E. Serum Protein Electrophoresis in wildtype C57Bl/6 is not expected to show a change in the protein fractions reported in this figure. The authors should consult with someone proficient at analyzing SPEPs (e.g., a hematologist or lab technician). The abnormality seen with age in C57Bl/6 mice is a very faint monoclonal band in the beta or, more commonly, the gamma globulins. It is not of sufficient size to increase the level of total globulins in the fraction. About 30% of 1 year old C57Bl/6 mice will have a faint monoclonal band detected on SPEP.

Response:

We thank Reviewer #3 for the insightful comment regarding the interpretation of SPEP results in C57Bl/6 mice. The serum protein electrophoresis analyses were performed and evaluated by a certified technician at a commercial diagnostic laboratory with expertise in SPEP interpretation.

Our objective was to assess overt monoclonal gammopathy consistent with multiple myeloma development rather than to detect subtle age-associated monoclonal bands. We acknowledge that faint monoclonal bands can occur in a proportion of aged C57Bl/6 mice and may not significantly alter total globulin fractions.

As the plasmacytoma induction model proved to be the most robust and informative system for mechanistic analyses, further investigation of subtle age-related SPEP alterations was not pursued as it was beyond the scope of this study and our expertise.

Table S12, S13, S14, S15. These bedpe files have a terminal column with numerical entries such as ;3/4;5/6. I suspect this is supposed to be left split reads/discordant reads; right split reads/discordant reads supporting the presence of the SV. It is not clear to me why all 4 numbers reported for each SV are identical, please check that this is correct. Furthermore, they appear to use a very low cutoff to report SV (3 reads supporting SV). This will be more sensitive to subclonal SV, but also introduces a lot of noise. I suspect a higher cutoff will provide a cleaner analysis, although they have convincingly made their point with their current analysis (if less than ideal).

We thank the reviewer for the careful evaluation of Tables S12–S15 and for pointing out the lack of clarity.

In the BEDPE files (SVDetect output), the relevant fields are as follows:

Field 7 – SVname: unique structural variant identifier (sequential ID + sample + SV type + annotation).

Field 8 – score: SVDetect weighted confidence score after filtering (range 0–1; higher values indicate stronger support).

Field 11 – NumberOfPairsAfterFiltering/previousNumberOfPairs: number of supporting discordant read pairs after filtering relative to the initial number detected (e.g., 25/30).

The filtering steps include strand filtering, order filtering, and insert size filtering. For clarity, we now report only the initial and final numbers of supporting reads. In many cases, the number of supporting reads remains unchanged after filtering because conservative read filtering had already been applied at the mapping stage. We have corrected the tables, introduced headers, and expanded their descriptions in the Supplementary Information to improve clarity.

Regarding the cutoff for SV detection, our approach was not optimized for exhaustive identification of high-confidence clonal driver rearrangements. Instead, the analytical pipeline was intentionally designed to capture a broad spectrum of statistically supported translocation events, including subclonal variants, in order to characterize the overall translocation landscape and sequence specificity associated with the DIS3^{G766R} mutation.

We additionally tested analyses using higher read-support cutoffs and did not observe significant differences in the overall conclusions. Therefore, we retained the more inclusive threshold to preserve sensitivity for subclonal events relevant to the mutator phenotype under investigation.

One of the IgH SV they report in the G766R Dis3 plasmacytomas has a well-supported breakpoint (67 reads) 170kb from Tnfrsf17 (Bcma), a gene that is sometimes similarly dysregulated in human myeloma. It would be nice if they can demonstrate over-expression of Tnfrsf17 in the sample with this translocation.

Response:

We appreciate this insightful suggestion and the potential link to human myeloma. The breakpoint located ~170 kb from Tnfrsf17 (Bcma) is indeed notable.

Unfortunately, the samples were collected exclusively for DNA-based analyses, and the animals were euthanized in accordance with ethical requirements at the time of collection, so no RNA material is available to assess Tnfrsf17 expression.

While evaluating expression would be informative, our study is focused on the mutational landscape and mechanistic consequences of DIS3^{G766R} dysfunction rather than on the oncogenic potential of individual structural variants. Therefore, this analysis falls beyond the scope of the current work.

I am unable to find the MYC SV shown in Figure S6E in Table S12. The SV in S6B,C,D,E do not overlap DRIPseq peaks as in Figure 4K and S6A. It is not clear how they support their argument.

Response:

The MYC SV shown in Figure S6C is the following:

```
chr11 109011984 109012059 chr15 61984826 61984901
SVid_7676_Het_57p_TRANSLOC;UNBAL 1 . . 14/14
```

Although this translocation reached a significant SVDetect confidence score and is supported by 14 high-quality discordant read pairs, it did not pass the additional filtering criteria we applied for breakpoints included in the meta-feature analysis.

The examples of translocations presented in Supplementary Figure 6 were selected as representative loci of potential biological relevance (e.g., immunoglobulin light chain variable regions, known biological target - Myc, and selected intergenic/non-coding regions). While not all breakpoints directly overlap DRIP-seq peaks, they display DRIP-seq or RNA-seq signal enrichment at or in close proximity to the translocation sites. This is in accordance with the meta-feature analysis shown in Fig. 4.

Finally, breakpoint mapping by mate-pair sequencing has limited resolution, and only a subset of mapped SVs define the exact DNA breakpoints. Therefore, absence of direct overlap does not exclude spatial proximity to R-loop-enriched regions.

I am unable to access the GSE155631. Be sure it contains the older mate-pair, and newer whole genome data. In my experience mouse whole genome and mate pair DNA sequencing data is submitted to SRA, not GEO.

We have confirmed GSE155631 accessibility (public as of 11.02.2026) and SRA (accession SRP275679, BioProject PRJNA650522). We have revised the data accessibility section to accommodate

Response:

We have submitted both the raw sequencing data to the SRA and the processed analysis results through GEO, taking advantage of the common submission process they offer to ensure that all accession numbers are cross-referenced between databases.

We have confirmed that GSE155631 is publicly accessible (as of 11 February 2026). The raw data are available in SRA under accession number SRP275679, associated with BioProject PRJNA650522.

We have revised the Data Availability section to explicitly include the SRA and BioProject accession numbers.